# Bacterial killing by complement requires membrane attack complex formation via surface-bound C5 convertases

Dani AC Heesterbeek[1,†], Bart W Bardoel[1,†], Edward S Parsons[2], Isabel Bennett[2], Maartje Ruyken[1], Dennis J Doorduijn[1], Ronald D Gorham Jr[1], Evelien TM Berends[1], Alice LB Pyne[2], Bart W Hoogenboom[2,3] ⓘD & Suzan HM Rooijakkers[1,*] ⓘD

## Abstract

The immune system kills bacteria by the formation of lytic membrane attack complexes (MACs), triggered when complement enzymes cleave C5. At present, it is not understood how the MAC perturbs the composite cell envelope of Gram-negative bacteria. Here, we show that the role of C5 convertase enzymes in MAC assembly extends beyond the cleavage of C5 into the MAC precursor C5b. Although purified MAC complexes generated from preassembled C5b6 perforate artificial lipid membranes and mammalian cells, these components lack bactericidal activity. In order to permeabilize both the bacterial outer and inner membrane and thus kill a bacterium, MACs need to be assembled locally by the C5 convertase enzymes. Our data indicate that C5b6 rapidly loses the capacity to form bactericidal pores; therefore, bacterial killing requires both *in situ* conversion of C5 and immediate insertion of C5b67 into the membrane. Using flow cytometry and atomic force microscopy, we show that local assembly of C5b6 at the bacterial surface is required for the efficient insertion of MAC pores into bacterial membranes. These studies provide basic molecular insights into MAC assembly and bacterial killing by the immune system.

**Keywords** atomic force microscopy; complement; convertase; Gram-negative bacteria; membrane attack complex
**Subject Categories** Membrane & Intracellular Transport; Microbiology, Virology & Host Pathogen Interaction
**The EMBO Journal (2019) 38: e99852**

## Introduction

Membrane attack complex (MAC) formation is an evolutionarily conserved immune mechanism to kill bacteria and altered self-cells.

It results from activation of the complement cascade (present in blood and most bodily fluids; Kang *et al*, 2009; Ricklin *et al*, 2010), when newly formed C5b6 complexes bind C7, C8, and multiple C9 molecules to build hetero-oligomeric MAC pores into target cell membranes. The MAC has an essential role in human immune protection against Gram-negative bacteria; this is evident from recurrent infections in patients lacking MAC activity due to genetic deficiencies (Ram *et al*, 2010; Turley *et al*, 2015) or due to treatment with complement-inhibitory drugs (Konar & Granoff, 2017; McNamara *et al*, 2017; Ricklin *et al*, 2017). Since MAC-dependent cell lysis can be specifically triggered via antibodies, this killing mechanism is also exploited for therapeutic development of antibodies that target cancer cells or drug-resistant bacterial infections (Szijártó *et al*, 2015; de Jong *et al*, 2016). Despite its crucial role in immunity, it is currently not understood how the MAC kills bacteria.

*In vivo*, the MAC is generated via an enzymatic chain reaction on the target cell surface (Ricklin *et al*, 2010; Berends *et al*, 2014). Following recognition of a foreign cell via antibodies or pattern recognition molecules, proteins of the complement system (Ricklin *et al*, 2010; Ugurlar *et al*, 2018) rapidly organize into a proteolytic cascade that eventually results in cleavage—by C5 convertase enzymes—of precursor C5 into the anaphylatoxin C5a and C5b (Gros *et al*, 2008; Ricklin *et al*, 2010); C5b initiates the assembly of the MAC (C5b-9; Hadders *et al*, 2012; Serna *et al*, 2016; Sharp *et al*, 2016; Bayly-Jones *et al*, 2017). Nascent C5b by itself is labile, but forms stable C5b6 complexes by rapid association with C6 (Cooper & Müller-Eberhard, 1970; Hadders *et al*, 2012). Next, C5b6 complexes bind C7, which changes conformation to expose a hydrophobic domain that renders the complex lipophilic (Preissner *et al*, 1985). Once C5b-7 is bound to the target membrane, its assembly with C8 and 18 copies of C9 results in the formation of the MAC (Bayly-Jones *et al*, 2017; preprint: Parsons *et al*, 2018). Recent *in vitro* structural studies (Serna *et al*, 2016; Sharp *et al*, 2016; Menny *et al*, 2018; preprint: Parsons *et al*, 2018) have revealed detailed information on how MAC proteins form toroid-shaped

1  Department of Medical Microbiology, University Medical Center Utrecht, Utrecht University, Utrecht, The Netherlands
2  London Centre for Nanotechnology, University College London, London, UK
3  Department of Physics and Astronomy, University College London, London, UK
   *Corresponding author. Tel: +31 88 75 504 86; E-mail: s.h.m.rooijakkers@umcutrecht.nl
   †These authors contributed equally to this work

pores with an inner diameter of 10 nm, spanning a single phospholipid (bilayer) membrane. Despite these structural insights, it remains unclear how these pores can kill Gram-negative bacteria. Since the cytoplasmic membrane of Gram-negative bacteria is physically protected by a peptidoglycan layer and an outer membrane (Silhavy *et al*, 2010), it is unclear how MAC pores, with a transmembrane region of < 10 nm (Serna *et al*, 2016; Sharp *et al*, 2016), can perturb this composite cell wall. Furthermore, it is not known if MAC can perturb both membranes and if other serum components, such as the peptidoglycan-degrading enzyme lysozyme (Wright & Levine, 1981), are required to kill a bacterium.

In this paper, we demonstrate that although the purified MAC components (C5b-9) form pores in artificial lipid membranes (Serna *et al*, 2016; Sharp *et al*, 2016), they require additional complement components to be bactericidal. Specifically, we find that MAC-dependent killing critically depends on the prior labeling of the bacterial surface with C5 convertase enzymes. Using novel membrane perturbation analyses and atomic force microscopy, we here show that *in situ* cleavage of C5 by convertases at the microbial surface and immediate insertion of newly formed C5b67 complexes into the membrane is essential for the MAC to perturb both bacterial membranes and to be bactericidal. Our data highlight a critical role for complement activation mechanisms at the cell surface to induce bacterial killing.

# Results

## MAC in serum perturbs both the outer and inner membrane of Gram-negative bacteria

To better understand bacterial killing by the MAC, we developed a flow cytometry-based approach that can distinguish between outer and inner membrane perforation in Gram-negative bacteria following exposure to human serum, which contains all complement proteins and has potent bactericidal activity against Gram-negative bacteria (Berends *et al*, 2014). Outer membrane integrity was monitored by measuring release of mCherry from the periplasmic space of genetically engineered *E. coli* MG1655 cells (Fig 1A and B). Inner membrane integrity was monitored by detecting release of cytosolic Green Fluorescent Protein (GFP) or by the influx of small molecule DNA dyes (Lebaron *et al*, 1998). Upon exposure of these cells to human serum, the periplasmic mCherry signal decreased in the entire population, indicating permeabilization of the outer membrane (Figs 1C and EV1A). Although cytosolic GFP signals remained constant, human serum induced effective passage of small DNA dyes (Figs 1C and EV1A).

By carefully titrating concentrations of serum (and thus complement components), we found that the influx of DNA dyes, but not mCherry release, strongly correlated with bacterial cell death (Figs 1C and EV1B). Also, in two wild-type *E. coli* strains and a clinical isolate of *Stenotrophomonas maltophilia* (*S. maltophilia*), we observed that serum-induced inner membrane disruption correlated with bacterial killing (Fig 1D and E). Both inner membrane disruption and killing by human serum fully relied on the presence of MAC components but not on the peptidoglycan-degrading lysozyme (Figs 1D and E, and EV1C). Taken together, these data suggest that MAC-mediated disruption of the inner membrane is an essential requirement for killing of Gram-negative bacteria in human serum.

## Purified MAC components lack the bactericidal activity of serum

To assess how the MAC can damage both membranes, we used purified MAC components instead of serum. Although nascent C5b is unstable, rapid association with C6 (Cooper & Müller-Eberhard, 1970; Hadders *et al*, 2012) leads to formation of stable C5b6 complexes that initiate the assembly of the MAC (C5b6-9; Hadders *et al*, 2012; Serna *et al*, 2016; Sharp *et al*, 2016; Bayly-Jones *et al*, 2017; Fig 2A). Such stable C5b6 complexes can be generated by activating C5 and C6 on activating surfaces (in C7-deficient serum) and subsequently purify released C5b6 from the supernatant (van den Berg, 2000). Upon incubation with C7, C8, and C9, such purified, preassembled C5b6 complexes can form MAC pores in various cell types. This protocol was recently used for structure determination of the MAC following its formation in liposomes (Serna *et al*, 2016; Menny *et al*, 2018). In concordance with literature, we also observed that preassembled C5b6 complexes can associate with proteins C7, C8, and C9 to assemble lytic MAC pores (henceforward denoted as $_{C5b6}$MACs) in liposomes (Fig EV2A; Sharp *et al*, 2016) and mammalian erythrocytes [human (Fig 2B) and rabbit (Fig EV2B (Lachmann & Thompson, 1970)].

However, when preassembled C5b6 triggers MAC formation on bacteria, these pores lack the bactericidal activity of human serum (Fig 2C), even at protein concentrations exceeding those in blood (Fig EV2C). Although $_{C5b6}$MAC is not bactericidal, it decreased mCherry signals in the entire population, indicating permeabilization of the outer membrane (Fig 2D). While $_{C5b6}$MAC pores perturb the bacterial outer membrane, they—unlike serum—lack the ability to damage the inner membrane of Gram-negative bacteria (Fig 2D and E). We conclude that $_{C5b6}$MAC lacks bactericidal activity in spite of its effectiveness in perturbing the outer membrane.

## Reconstituting bactericidal MAC assembly via surface-bound C5 convertases

To identify which additional factors in serum are needed to form bactericidal MAC pores, we next attempted to more closely mimic *in vivo* MAC assembly via cell-bound C5 convertases (Figs 3A and EV3A; Berends *et al*, 2014). In serum, antibodies or pattern recognition molecules specifically drive the deposition of C5 convertases onto the target cell surface (Gros *et al*, 2008; Ricklin *et al*, 2010; Berends *et al*, 2014). This occurs in a step-wise manner (Fig EV3A): firstly, all three recognition pathways deposit C3 convertase enzymes that cleave protein C3 into C3b, which covalently attaches to the cell surface via a reactive thioester. At high densities of surface-bound C3b, C3 convertases associate with deposited C3b to form a C5 convertase. Although their surface-specific nature and covalent attachment make it technically challenging to generate C5 convertases in a purified manner, we here show that pre-incubation of bacteria with human serum devoid of C5 (ΔC5 serum) results in functionally active (mainly alternative pathway) C5 convertases on the bacterial surface (Fig EV3B–D).

When such convertase-labeled bacteria were next washed and then incubated with uncleaved C5 and components C6, C7, C8, and C9, we found that the resulting MAC pores (denoted as Conv-MAC) effectively killed bacteria (Fig 3B) in a C5 dose-dependent manner (Fig 3C). As is the case *in vivo*, C5 convertases here were essential

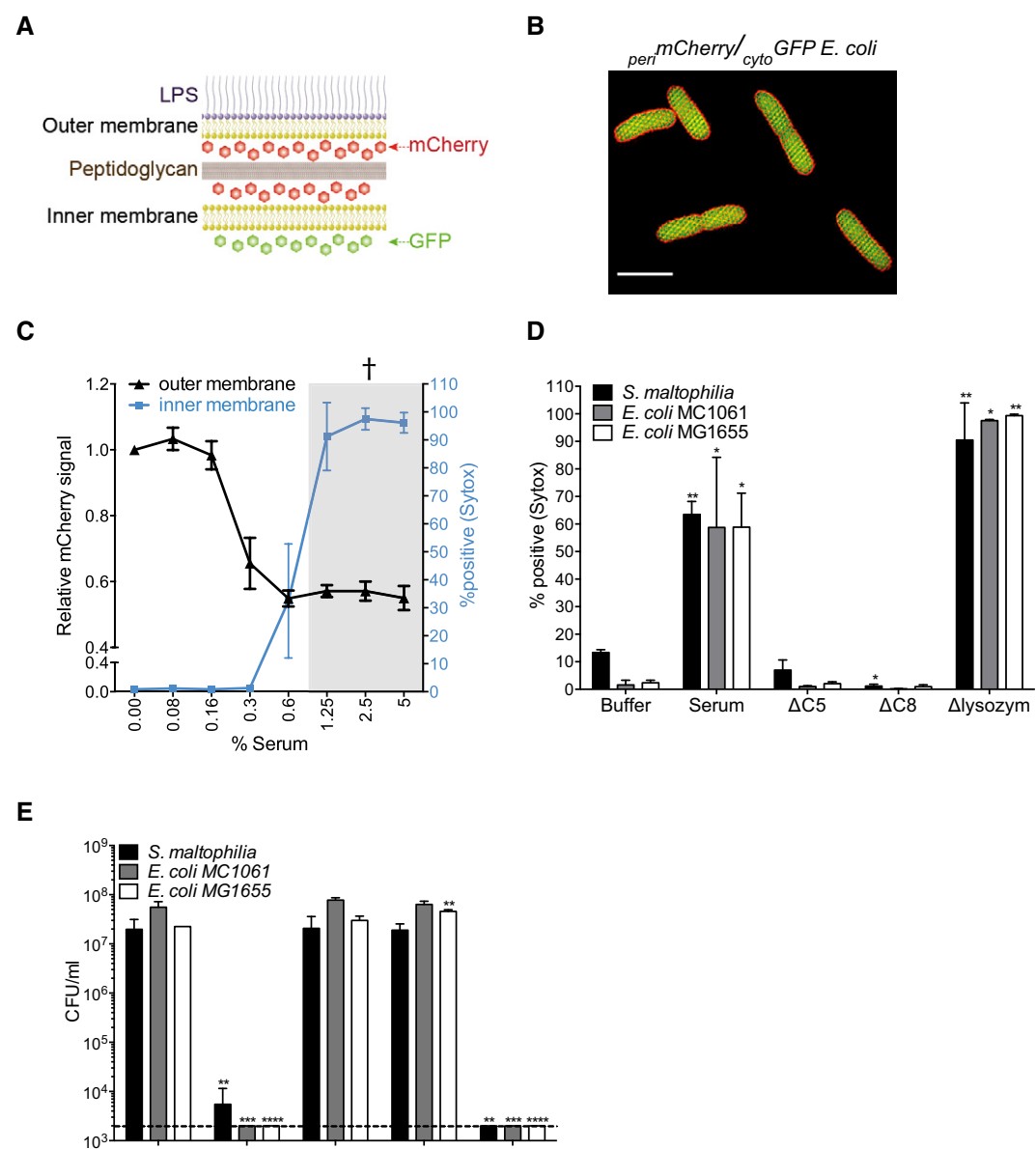

**Figure 1.  MAC in serum perturbs both the outer and inner membrane of Gram-negative bacteria.**

A      Schematic representation of engineered _peri_mCherry/_cyto_GFP *E. coli* cells that express mCherry in the periplasmic space (between the outer and inner membrane) and GFP in the cytosol.

B      Structured illumination microscopy image of _peri_mCherry/_cyto_GFP *E. coli* confirming localization of mCherry (red) in the periplasm and GFP (green) in the cytosol. Scale bar = 3 μm.

C      Outer membrane damage (mCherry intensity) and inner membrane damage (% Sytox positive) of _peri_mCherry/_cyto_GFP *E. coli* bacteria exposed to (different concentrations of) human serum. Inner membrane damage correlates with killing (samples where bacteria are killed are indicated with gray shadings and a cross, see CFU data in Fig EV1B).

D, E    (D) Serum-induced inner membrane damage (% Sytox positive) and (E) killing (CFU/ml) of different Gram-negative strains depends on MAC components C5 and C8, but not on lysozyme (10% serum). Dotted line represents the detection limit of the assay.

Data information: The cfu/ml (E) and Sytox measurements (D) of "Buffer", "Serum", "ΔC5", "ΔC8", "Δlysozym", and "_C5b6_MAC" were all generated from the same experiment. (C–E) Data represent mean ± SD of 3 independent experiments. (D, E) Statistical analysis was done using a ratio paired two-tailed *t*-test and displayed only when significant as *$P ≤ 0.05$, **$P ≤ 0.01$, ***$P ≤ 0.001$, or ****$P ≤ 0.0001$.

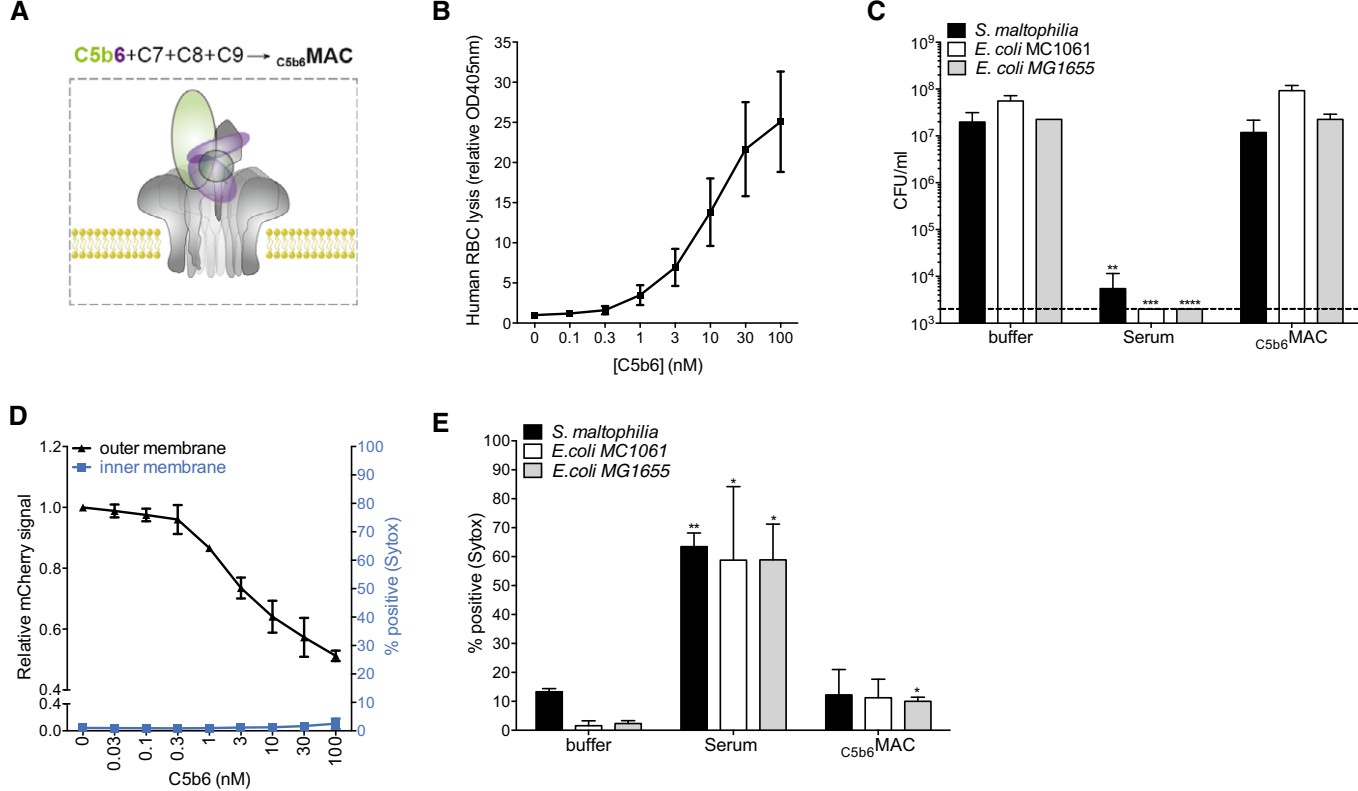

**Figure 2. Purified MAC components lack the bactericidal activity of serum.**

A  Purified MAC (denoted as $_{C5b6}$MAC) can be formed by mixing preassembled C5b6 complexes with C7, C8, and C9.

B  Lysis of human erythrocytes after exposure to a concentration range of preassembled C5b6 in the presence of 100 nM C7. After washing, erythrocytes were exposed to 20 nM C8 and 100 nM C9 for 30 min after which the OD405 nm of the supernatant was measured.

C  Bacterial viability of three Gram-negative strains after exposure to buffer, 10% human serum or $_{C5b6}$MAC. Buffer and serum conditions are the same as Fig 1E.

D  Permeabilization of the outer, but not inner membrane of $_{peri}$mCherry/$_{cyto}$GFP *E. coli* cells exposed to $_{C5b6}$MAC (different concentrations of C5b6 with fixed concentrations of C7-C9).

E  Inner membrane damage of three Gram-negative strains exposed to buffer, 10% serum or $_{C5b6}$MAC. Buffer and serum conditions are the same as Fig 1D.

Data information: The cfu/ml (C) and Sytox measurements (E) of "Buffer", "Serum", "ΔC5", "ΔC8", and "Δlysozym", and "$_{C5b6}$MAC" were all generated from the same experiment. (B–E) Data represent mean ± SD of 3 independent experiments. (C, E) Statistical analysis was done using a ratio paired two-tailed *t*-test and displayed only when significant as *$P \leq 0.05$, **$P \leq 0.01$, ***$P \leq 0.001$, or ****$P \leq 0.0001$. Normal concentrations of MAC proteins in 100% human serum are ± 375 nM C5, 550 nM C6, 600 nM C7, 350 nM C8, and 900 nM C9.

for bacterial killing, since no killing was observed when bacteria were pre-incubated with convertase-negative serum (heat-inactivated ΔC5 serum (Fig 3C), which lacks the capacity to deposit C3b (Fig EV3C) and to convert C5 (Fig EV3D). In addition, bacterial killing was fully inhibited when convertase formation was blocked by compstatin (Sahu *et al*, 1996; Fig 3D), which specifically inhibits surface deposition of C3b (Fig EV3C) and formation of functional C5 convertases (Fig EV3D). The here observed bacterial killing was MAC-specific, as it required the presence of C5-C9 to be fully effective (Fig 3D). In the presence of C5-C8 alone (no C9), bacterial killing was much less effective but not insignificant (Fig 3D), supporting previous reports on bactericidal effects of C5b-8 in the absence of C9 (O'Hara *et al*, 2001). Similar to these results based on alternative pathway C5 convertases, specific labeling of bacteria with classical/lectin pathway C5 convertases also led to bacterial cell death upon incubation with C5-C9 (Fig 3E). Altogether, these data demonstrate that purified MAC can kill bacteria when its assembly is driven by cell-bound C5 convertases.

### MAC assembly via surface-bound C5 convertases leads to inner membrane damage

Given that MAC assembly via surface-bound C5 convertases results in bacterial cell death, we assessed whether it also results in inner membrane damage. To study this, $_{peri}$mCherry/$_{cyto}$GFP bacteria were labeled with C5 convertases of the alternative pathway, washed, and next incubated with components C5-C9, as described above. In concordance with its bactericidal activity, we found that convertase-driven MAC assembly perturbed both the outer and inner membrane of $_{peri}$mCherry/$_{cyto}$GFP *E. coli*, in a C5 dose-dependent manner (Fig 4A). In accordance with the killing data in Fig 3C and D, we observed that inhibition of C5 convertase formation via heat inactivation or by adding compstatin prevented MAC-mediated inner membrane damage (Fig 4B). Furthermore, C5 convertase-generated MAC pores (Conv-MAC) also induced inner membrane damage in wild-type *E. coli* and *S. maltophilia* strains (Fig 4C). Consistent with the flow cytometry results, confocal microscopy

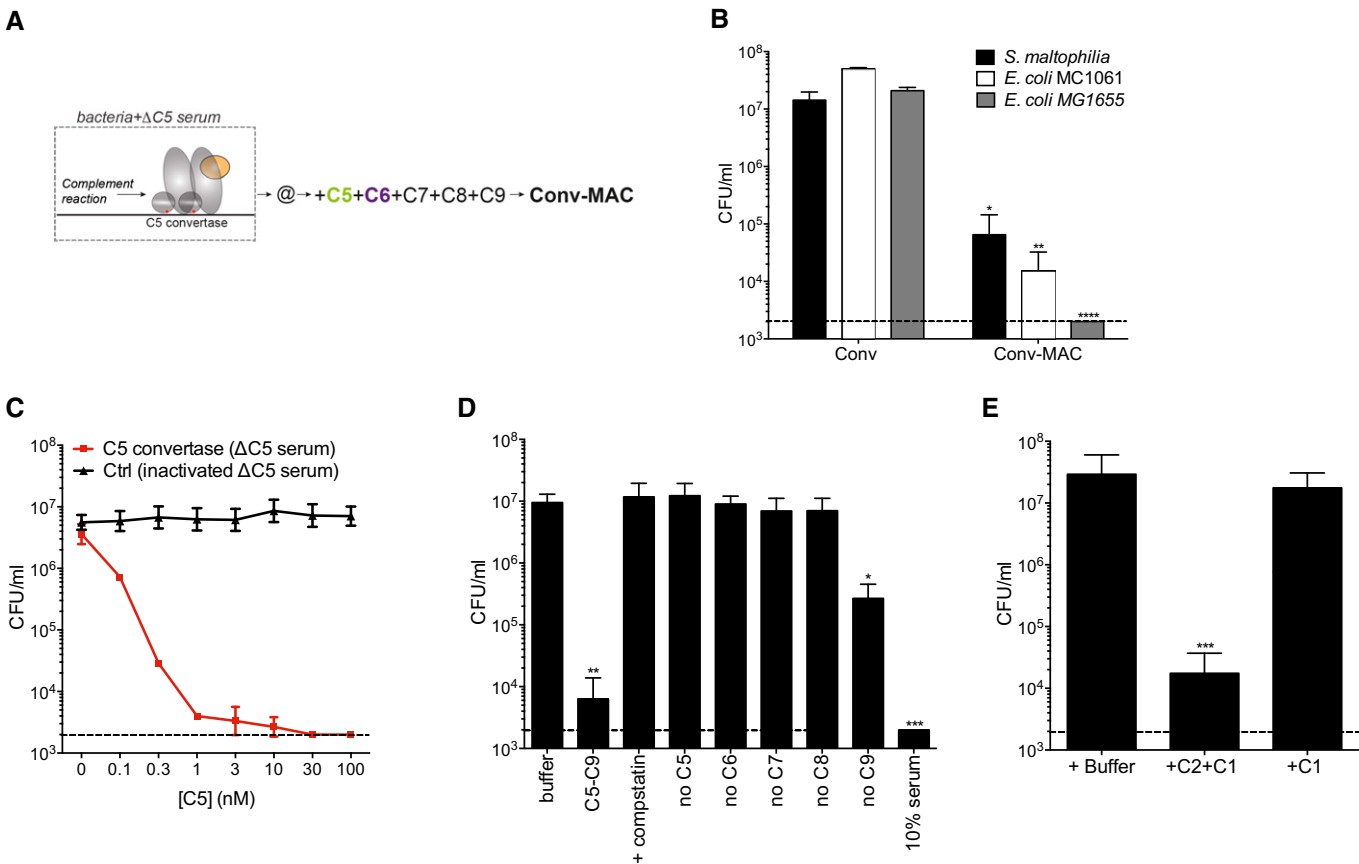

**Figure 3.  Reconstituting bactericidal MAC assembly via surface-bound C5 convertases.**

A   Schematic overview for Conv-MAC formation. Bacteria were labeled with C5 convertases by pre-incubation with C5-deficient serum (Fig EV3). Following a washing step (@), convertase-labeled bacteria were incubated with uncleaved C5, C6, C7, C8, and C9 (termed "Conv-MAC").

B   Bacterial viability of convertase-labeled bacterial strains exposed to buffer (Conv) or C5-C9 (Conv-MAC).

C   Bacterial viability of convertase-labeled *E. coli* MG1655 exposed to a concentration range of C5 in the presence of 100 nM C6, 100 nM C7, 20 nM C8, and 100 nM C9. "Ctrl" indicates bacteria that are pretreated with heat-inactivated ΔC5 serum. Dotted line represents the detection limit of the assay.

D   Bacterial viability of convertase-labeled *E. coli* MG1655 exposed to C5-C9 or conditions lacking one MAC component. As an extra control, convertase formation was blocked during ΔC5 serum incubation by adding 5 μM compstatin.

E   Bacterial viability of *E. coli* MG1655 exposed to FB depleted serum in the presence of 20 μg/ml OmCI (to deposit C4b and C3b without Bb). After washing, bacteria were exposed to C5-C9 in the presence or absence of C1 and C2 (to generate classical pathway C5 convertases, C4b2aC3b).

Data information: (B–E) Data represent mean ± SD of 3 independent experiments. (B, D, E) Statistical analysis was done using a ratio paired two-tailed *t*-test and displayed only when significant as *$P \leq 0.05$, **$P \leq 0.01$, ***$P \leq 0.001$, or ****$P \leq 0.0001$.

---

further confirmed that the combination of surface-bound C5 convertases and MAC induces inner membrane damage in bacteria (Fig 4D). In conclusion, when the MAC is assembled from purified C5-C9 by surface-bound convertases, these pores trigger inner membrane damage and subsequent bacterial killing.

## Local assembly of C5b6 by surface-bound C5 convertases is required for bacterial killing

Having established a protocol to generate bactericidal MACs under semi-purified conditions, we next investigated the difference between the non-bactericidal MACs formed by preassembled C5b6 and fully functional, convertase-generated MACs formed from C5 and C6. To this end, we labeled bacteria with C5 convertases as described above and subsequently generated MACs by incubation

with preassembled C5b6 and C7-C9 (Conv-$_{C5b6}$MAC) or by incubation with uncleaved C5 and C6 and C7-C9 (Conv-MAC; Fig 5A). In this direct comparison, only the Conv-MAC killed bacteria (Fig 5B) and triggered inner membrane damage (Fig 5C), whereas both Conv-$_{C5b6}$MAC and Conv-MAC triggered outer membrane damage (Fig 5D). These results indicate that the role of C5 convertases extends beyond the cleavage of C5 into C5a and C5b: In particular, the C5 convertase should be present on the cell surface and locally assemble C5b6 to generate bactericidal pores. Interestingly, and fully consistent with the lytic function of $_{C5b6}$MAC on liposomes (Fig EV2A) and mammalian erythrocytes (Figs 2B and EV2B), local assembly of C5b6 by surface-bound C5 convertases was not essential to kill human cells (Fig EV4; note that this experiment was performed on human (HAP1) cells deficient in complement regulators CD46, CD55, CD59 (Thielen *et al*, 2018)

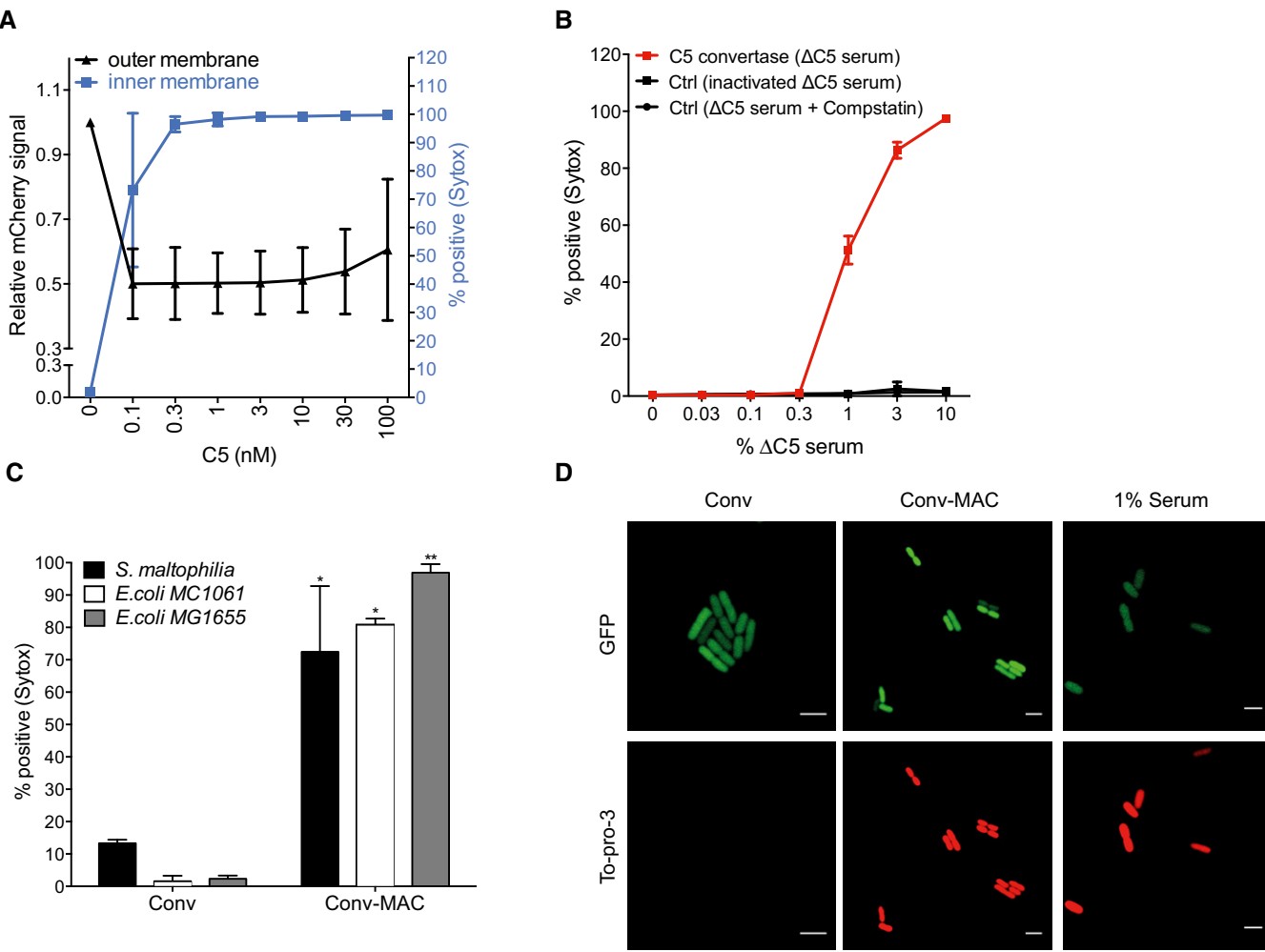

**Figure 4. MAC assembly via surface-bound C5 convertases leads to inner membrane damage.**

A   Outer membrane damage (mCherry intensity) and inner membrane damage (% Sytox positive) of convertase-labeled $_{peri}$mCherry/$_{cyto}$GFP *E. coli* cells incubated with a concentration range of C5 and fixed concentrations of C6-C9.

B   Inner membrane damage of $_{peri}$mCherry/$_{cyto}$GFP *E. coli* exposed to a concentration range of ΔC5 serum and, after washing, to C5-C9. As controls, bacteria were incubated with heat-inactivated ΔC5 serum or 5 μM compstatin was added to the ΔC5 serum to block C3b deposition.

C   Inner membrane damage of three different convertase-labeled bacteria exposed to buffer (Conv) or C5-C9 (Conv-MAC).

D   Confocal microscopy images of convertase-labeled $_{peri}$mCherry/$_{cyto}$GFP *E. coli* exposed to buffer (Conv) or C5-C9 (Conv-MAC). Unlabeled bacteria exposed to 1% serum served as control. Green = GFP, red = To-pro-3 DNA dye. Scale bars = 3 μm.

Data information: (A–C) Data represent mean ± SD of 3 independent experiments. (C) Statistical analysis was done using a ratio paired two-tailed *t*-test and displayed only when significant as *$P \leq 0.05$ or **$P \leq 0.01$.

that, if present, would prevent any C3b or MAC deposition on the cell surface).

Given that $_{C5b6}$MAC (Fig 2D) and Conv-$_{C5b6}$MAC (Fig 5D) pores both perturb the bacterial outer membrane, but not the inner membrane, the local assembly of C5b6 via surface-bound C5 convertases seems particularly important to generate pores that can damage the complete, composite cell envelope of a Gram-negative bacterium.

**C5b6 rapidly loses the capacity to form bactericidal pores**

To understand why local formation of C5b6 is required, we next focused on the early assembly steps of the MAC, involving C5b, C6, and C7. In the experiments described above, bactericidal pores

(Conv-MAC) were generated when convertase-labeled were simultaneously incubated with C5-C9 (Fig 4) [or when directly compared to $_{C5b6}$MAC, with C5-C7 and after washing C8 and C9 (Fig 5)]. Next, we studied whether we could introduce a washing step between the formation of C5b6 and the addition of C7. Interestingly, we found that washing after C5b6 assembly did not affect the formation of MAC pores that permeabilize the outer membrane (Fig 6A), but strongly blocked formation of MAC pores that trigger inner membrane damage (Fig 6B). In contrast, when C5b6 was formed in the presence of C7, we observed that washing did not affect the ability of MACs to permeabilize the outer and inner membrane (Fig 6A and B). This suggests that locally assembled C5b6 requires immediate insertion into the membrane via C7. In a different experimental

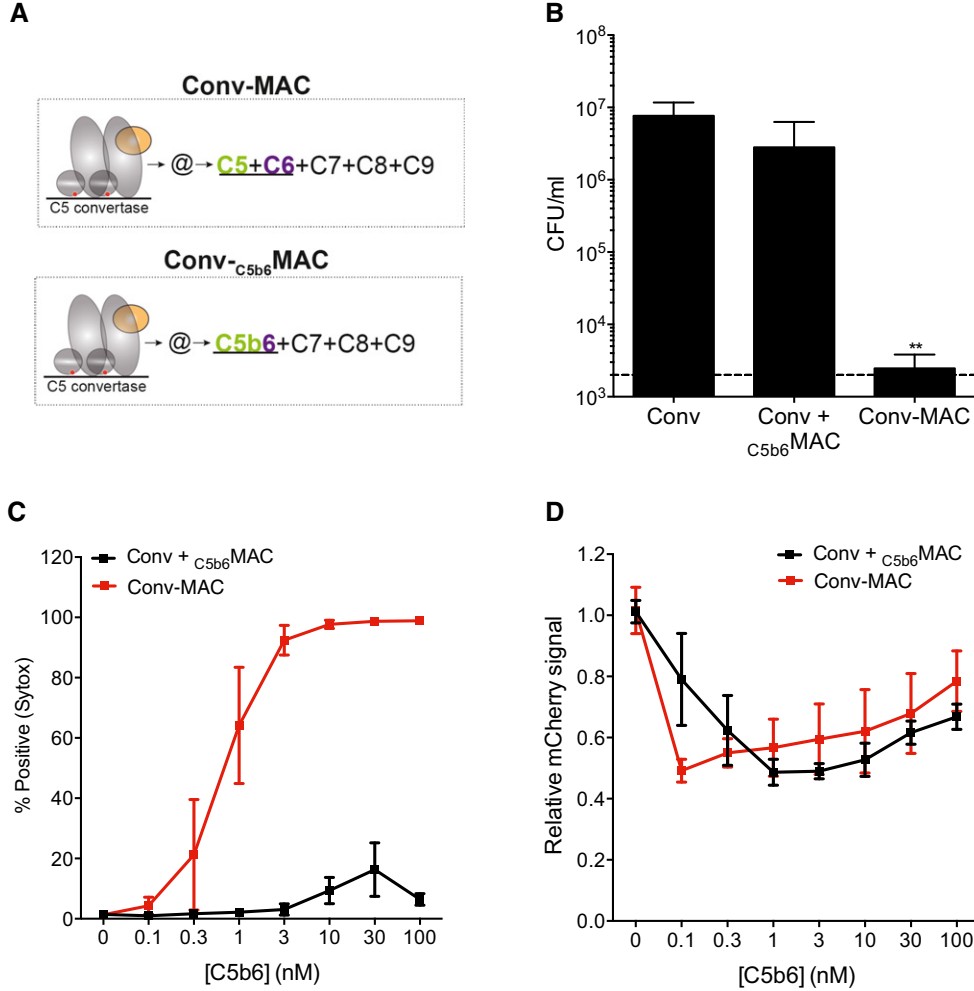

**Figure 5.  Local assembly of C5b6 by surface-bound C5 convertases is required for killing.**

A    Schematic overview of MAC assembly on convertase-labeled bacteria by C5b6 that is locally generated by incubation with C5 and C6 (top) or by preassembled C5b6 (bottom).

B    Bacterial viability of convertase-labeled *E. coli* MG1655 exposed to Buffer (Conv), preassembled C5b6 (Conv + $_{C5b6}$MAC) or a mixture of C5 and C6 (Conv-MAC), in the presence of C7, C8, and C9. Dotted line represents the detection limit of the assay.

C, D    (C) Inner membrane damage (% Sytox positive) and (D) outer membrane damage (mCherry) of convertase-labeled $_{peri}$mCherry/$_{cyto}$GFP *E. coli* exposed to a concentration range of preassembled C5b6 or a mixture of C5 and C6, in the presence of 100 nM C7. After washing, bacteria were exposed to 20 nM C8 and 100 nM C9.

Data information: (B–D) Data represent mean ± SD of 3 independent experiments. (B) Statistical analysis was done using a ratio paired two-tailed *t*-test and displayed only when significant as **$P \leq 0.01$.

set-up, we did not wash the bacteria following incubation with C5 and C6, but instead blocked the generation of new C5b6 molecules using Eculizumab, a clinically approved C5 inhibitor that blocks recognition and cleavage of C5 by the convertase (Rother *et al*, 2007). We observed that Eculizumab blocked formation of bactericidal MAC pores when it was added after C5b6 assembly but before the addition of C7 (Fig 6A and B). In contrast, Eculizumab could not prevent bactericidal MAC formation when added after the addition of C5, C6, and C7 (Fig 6A and B). These data show that although C5b6 complexes that are formed in the absence of C7 can form pores that damage the outer membrane, these complexes rapidly lose the ability to form bactericidal pores. In contrast, when newly generated C5b6 can immediately bind C7 at the bacterial surface, these complexes

can form a bactericidal MAC. Together with the fact that purified, preassembled C5b6 complexes lack the capacity to form bactericidal MAC pores, these data indicate a previously unrecognized short-lived ability of C5b6 to induce bacterial killing and further explains why local assembly of C5b6 by C5 convertases on the target surface is so crucial for bacterial killing via the MAC.

### Inner membrane damage is driven by MAC assembly at the outer membrane

Next, we investigated how bactericidal MAC pores perturb the bacterial cell envelope. Specifically, we analyzed whether inner membrane damage results from MAC formation in the outer or inner membrane.

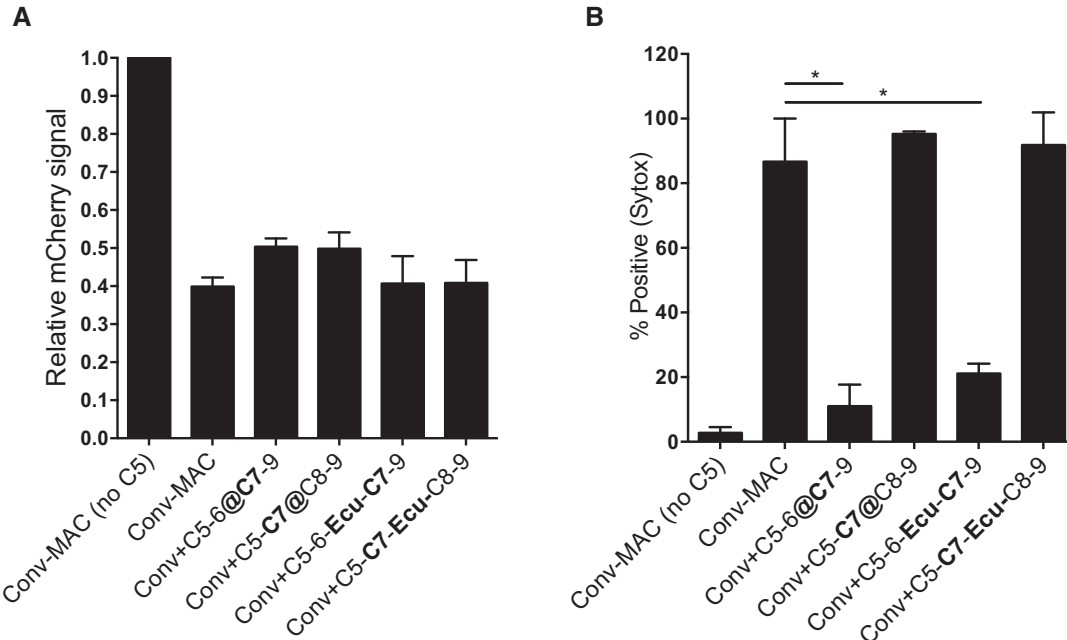

**Figure 6. C5b6 rapidly loses the capacity to form bactericidal pores.**

A, B  Step-wise assembly of MAC on convertase-labeled bacteria. Convertase-labeled bacteria were incubated with C5/C6 or C5/C6/C7 for 15 min, and subsequently washed (@) or treated with 10 µg/ml Eculizumab (Ecu). Then, the remaining MAC components (C7-9 for C5/C6 or C8-9 for C5/C6/C7, respectively) were added to the incubation mixture. In the control conditions (Conv-MAC), the remaining MAC components were added to the incubation mixture without washing or adding an inhibitor. (A) Outer membrane damage (mCherry) and (B) inner membrane damage (% Sytox positive) were determined.

Data information: (A-B) Data represent mean ± SD of 3 independent experiments. Statistical analysis was done using a ratio paired two-tailed *t*-test in which the test conditions were compared to Conv-MAC and displayed only when significant as *$P \leq 0.05$.

We performed step-wise assembly of MAC pores on convertase-labeled bacteria and assessed damage of both membranes at each step. First of all, we observed that the labeling of bacteria with convertases, and subsequent assembly of C5b-7 (in absence of C8 and C9), did not affect permeability of either membrane (Fig 7A). Convertase-driven formation of C5b-8 allowed some passage of small molecule DNA dyes (Sytox, Fig 7A) but not of proteins (mCherry; Fig 7A). As shown before, the formation of a full MAC pore (C5b-9) is required to effectively permeabilize both membranes (Fig 7A). To further disentangle the effect of MAC formation on the outer and inner membrane, we introduced a washing step after the incubation with C5-C8. At this stage, the outer membrane did not show significant permeability for proteins (mCherry, Fig 7A), leading us to conclude that all remaining C5b-8 must be bound to the outer membrane surface only. Intriguingly, subsequent incubation of (washed) bacteria with C9 led to both outer and inner membrane permeabilization (Fig 7A and B). This suggests that assembly of bactericidal MAC pores takes place in the bacterial outer membrane only and that destabilization of the inner membrane is of a different nature than outer membrane permeabilization, not requiring the step-wise assembly of new convertases or C5b-9 pores.

**Local formation of C5b6 is required for efficient insertion of MAC pores into the outer membrane**

Since the above experiments indicate that formation of bactericidal MAC pores mainly takes place on the outer membrane, we more closely analyzed pore formation on the bacterial surface. First, we quantified the total number of MAC assemblies on convertase-labeled bacteria using flow cytometry (Fig 8A). We compared C9-Cy3 incorporation on convertase-labeled bacteria when MAC formation was triggered via uncleaved C5 and C6 (Conv-MAC) or preassembled C5b6 (Conv-$_{C5b6}$MAC; solid lines in Fig 8A). The dose–response curves indicate that MAC formation was up to ~3-fold more efficient with locally formed C5b6 (Conv-MAC), although the difference was absent at higher (100 nM) concentrations of C5b6 (Fig 8A). This is in apparent contradiction with the vast differences in inner membrane damage between Conv-MAC and Conv-$_{C5b6}$MAC at these concentrations (Fig 5C). Since these differences could not be explained by a different distribution of Conv-$_{C5b6}$MAC and Conv-MAC pores at the cell surface either (Fig 8B), we next tested how well Conv-$_{C5b6}$MAC or Conv-MAC pores are inserted into the bacterial membrane by measuring their resistance to trypsin after MAC formation. Trypsin is commonly used as a shaving method to determine surface exposure of membrane-associated proteins (Moskovich & Fishelson, 2007; Besingi & Clark, 2015). While trypsin treatment effectively reduced the amount of C9-Cy3 incorporation for the preparations with preassembled C5b6, it had no effect on MAC pores generated via C5 and C6 (Fig 8A). Altogether, these data indicate that the $_{C5b6}$MACs are less well attached to and therefore presumably less well inserted into the bacterial membrane than Conv-MACs.

Next, we visualized MAC assembly in the outer membrane using atomic force microscopy (AFM) on live bacteria. Immobilized, untreated *E. coli* cells appeared as smooth rods (Fig 8C), which at

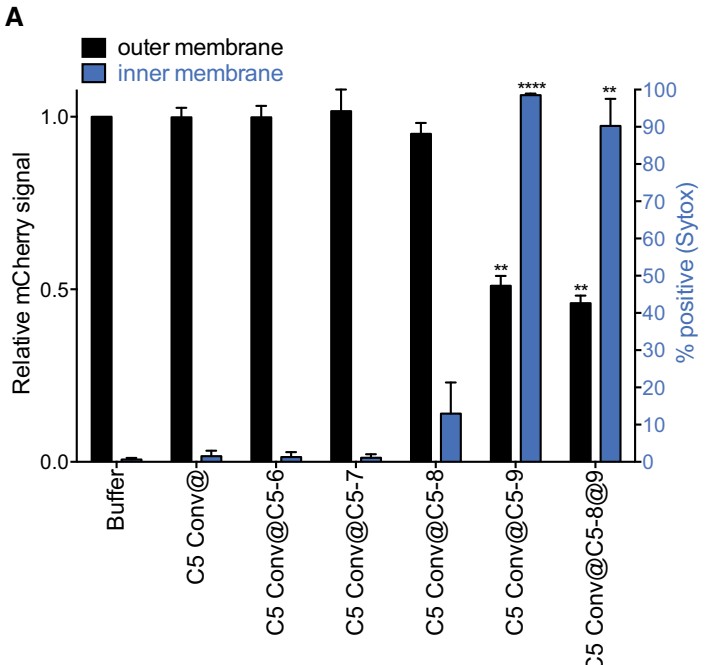

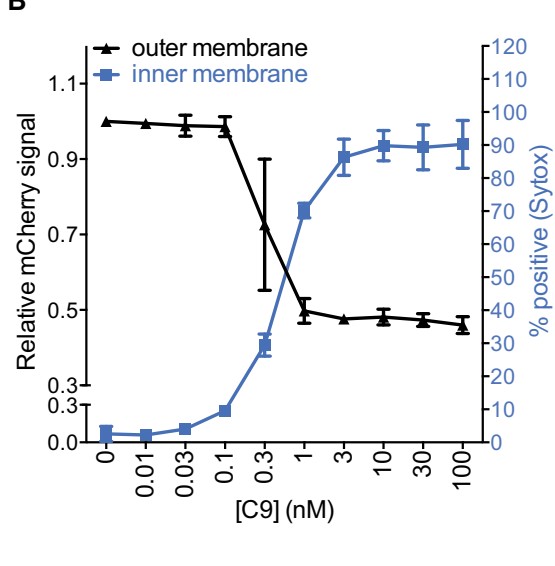

**Figure 7.  Inner membrane damage is driven by MAC assembly at the outer membrane.**

A   Outer and inner membrane damage of convertase-labeled bacteria exposed to different combinations of MAC components. "@" indicates a washing step.

B   Outer and inner membrane damage of convertase-labeled bacteria exposed to C5-C8 and after washing, to a concentration range of C9.

Data information: Data represent mean ± SD of 3 independent experiments. (A) Statistical analysis was done using a one-way ANOVA and displayed only when significant as **$P \leq 0.01$ or ****$P \leq 0.0001$.

high resolution showed densely packed, ~7-nm-wide porins, characteristic of the bacterial outer membrane (Yamashita *et al*, 2012). By contrast, following exposure of convertase-labeled bacteria to C5-C9 (Conv-MAC), the bacterial surface was covered with nanometer-scale protrusions that at higher resolution appeared as 10 ± 2 nm high and 17 ± 2 nm (peak to peak) wide pores, including—depending on AFM resolution—signatures of a C5b stalk extending upwards from the pore (Fig 8C). These dimensions and appearance are consistent with cryo-EM maps of MACs built up from preassembled C5b6 on liposomes (Serna *et al*, 2016; Sharp *et al*, 2016). At the surface of the immobilized, convertase-labeled bacteria, Conv-MAC pores were further accentuated in the phase image (see Materials and Methods), which provides an alternative means to differentiate MAC pores from the underlying bacterial surface since it is sensitive to the local material properties (Figs 8D and EV5A). However, when MACs were formed from preassembled C5b6 on convertase-labeled bacteria (Conv-$_{C5b6}$MAC), it became extremely challenging to discern pore structures at the bacterial surface (Figs 8D and EV5B and C). This is consistent with previous AFM experiments on related pore-forming proteins (Leung *et al*, 2014, 2017), in which inserted pores were readily detected on supported lipid bilayers, but mobile, non-inserted pores were harder to resolve due to the invasiveness of the AFM measurement and/or insufficient temporal resolution. Hence, the trypsin shaving and AFM results could be explained by inefficient insertion of Conv-$_{C5b6}$MACs into the membrane, implying that local assembly of C5b6 by surface-bound convertases is essential for priming the efficient insertion of MAC pores into bacterial membranes.

Finally, to further validate this explanation, we more closely analyzed the efficiency by which Conv-$_{C5b6}$MAC and Conv-MAC pores damage the bacterial outer membrane. At high (0.3 nM) C8 concentrations, it appeared that Conv-$_{C5b6}$MAC and Conv-MAC pores are equally efficient in inducing leakage of mCherry through the outer membrane (Fig EV5D), consistent with the findings in Fig 5D. However, by carefully titrating the concentration of C8, we observed that less pores (> 100-fold) are needed to induce maximum mCherry leakage in the conditions of locally assembled MAC (Conv-MAC) compared to preassembled Conv-$_{C5b6}$MAC (Fig EV5D). Together with the trypsin shaving and AFM results, these data indicate that local assembly of MAC pores triggers more efficient membrane insertion and subsequently more effective damage to the bacterial outer membrane. Future experiments are required to determine whether subsequent destabilization of the inner membrane (Fig EV5E) results directly from a more extensively damaged outer membrane or whether other mechanisms are at play to kill the cell.

## Discussion

Although recent *in vitro* structural studies (Dudkina *et al*, 2016; Serna *et al*, 2016; Sharp *et al*, 2016; Menny *et al*, 2018) have significantly advanced our understanding of MAC formation in liposomes, it is still not understood how these pores can damage the composite envelope of Gram-negative bacteria. Here, we reveal that the assembly and insertion of MAC pores into bacterial membranes differ from liposomes and erythrocytes. While MAC pores generated from

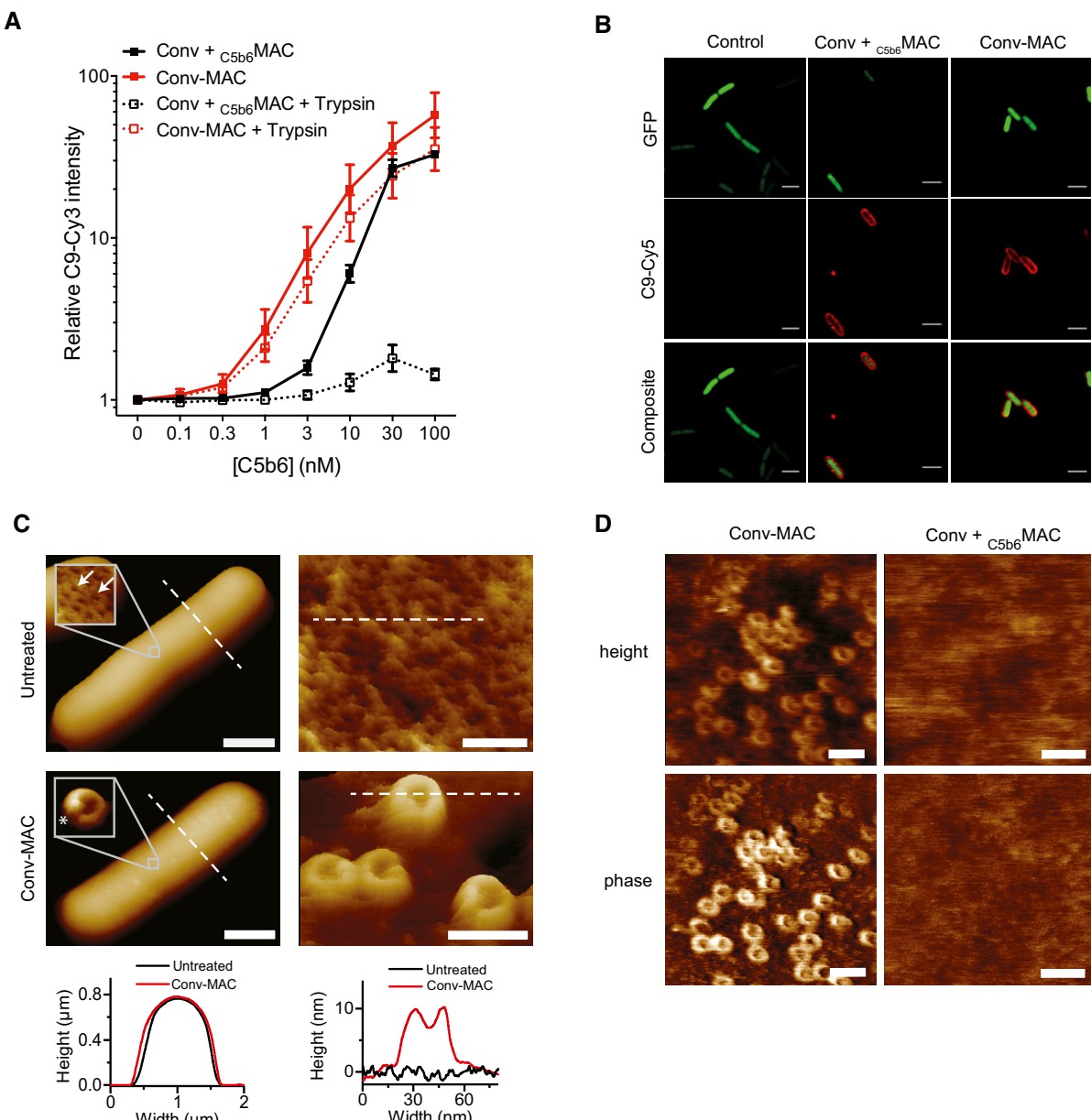

**Figure 8.  Local formation of C5b6 is required for efficient insertion of MAC pores into the outer membrane.**

Surface-bound MAC pores were quantified by flow cytometry or confocal microscopy.

A    Convertase-labeled bacteria were exposed to a concentration range of either preassembled C5b6 (C5b6MAC) or a mixture of C5 and C6 (Conv-MAC), in the presence of 100 nM C7. After washing, 20 nM C8 and 100 nM C9-Cy3 were added. Controls at 0 nM C5b6 or C5-C6 confirm that the detected C9-Cy3 deposition is specifically related to MAC formation (solid lines). Proper insertion of pores was assessed by a previously described shaving method with trypsin (Moskovich & Fishelson, 2007). Bacteria were first incubated with MAC components for 30 min and subsequently treated with 20 μg/ml trypsin for 15 min at 37°C (dotted lines).

B    Convertase-labeled perimCherry/cytoGFP bacteria (Green) exposed to C5b6MAC or Conv-MAC. Conditions were similar to those in (A); however, C9-Cy5 was used to detect MAC pores (Red). 100 nM of C5 and C6 or C5b6 was used in combination with 100 nM C7, 20 nM C8, and 100 nM C9-Cy5. Conv + C5b6MAC and Conv-MAC images were taken in separate experiments in which laser settings were adjusted to the staining intensity of C9-Cy5 to properly visualize pore distribution. Scale bars = 3 μm.

C, D    Atomic force microscopy analysis of *E. coli* BL21 and MG1655 immobilized using the Poly-L-Lysine protocol. (C) Entire bacteria and high-resolution comparisons of untreated and convertase-labeled *E. coli* BL21 exposed to C5-C9 (Conv-MAC) for 10 min. Scale bars: 800 nm (left) and 30 nm (right). Height scales: 1 μm (left), 8 nm (top right), 22 nm (bottom right). Width of magnification boxes: 42 nm, height scales: 8 nm (top) and 13 nm (bottom). Arrows highlight *E. coli* porin structures; an asterisk highlights the C5b-7 stalk. Height profiles (bottom) are shown for the white dashed lines in the images. (D) Atomic force microscopy (height and phase images) of convertase-labeled *E. coli* MG1655 exposed to a mixture of C5 and C6 (Conv-MAC) or preassembled C5b6 (Conv + C5b6MAC), in the presence of C7, C8, C9, FB, and FD. Images were generated in the same experiment. Scale bars: 50 nm. Height scales: 15 nm. This figure and three other replicates are included in Fig EV5B and C.

Data information: (A) Data represent mean ± SD of 3 independent experiments.

preassembled C5b6 can efficiently perforate single membrane particles, these MACs lack bactericidal activity. In order to kill a Gram-negative bacterium, MACs need to be assembled locally by cell-bound C5 convertase enzymes. Our data indicate that both the *in situ* conversion of C5 by surface-bound convertases and immediate association of C5b with C6 and C7 are needed to guide proper insertion of bactericidal MAC pores.

Our data imply that the C5b6 complex has a hitherto unrecognized limitation in its ability to induce bacterial killing. The purified C5b6 complex used in this study was generated by cleaving C5 in the presence of C6, but in the absence of C7, on an activating surface. Subsequently, the released C5b6 complexes were purified from the solution (van den Berg, 2000). Numerous reports have shown that these "preassembled" C5b6 complexes can efficiently form MAC pores in eukaryotic membranes (Iida *et al*, 1991) and liposomes (Michaels *et al*, 1976; Hu *et al*, 1981; Serna *et al*, 2016; Sharp *et al*, 2016; Menny *et al*, 2018; preprint: Parsons *et al*, 2018). We here found that the same preassembled C5b6 complexes lack the capacity to generate MACs that kill bacteria. In contrast, C5b6 that is formed locally by surface-bound C5 convertases efficiently forms bactericidal MAC pores. The fact that locally assembled C5b6 rapidly loses the ability to form bactericidal pores (Fig 6) suggests that C5b6 somehow becomes inactivated. Although the exact molecular mechanism for C5b6 inactivation is yet unknown, we here propose several possible explanations. First, cleavage of C5 by C5 convertases (hypothetical model in Fig 9A) results in a major conformational change in which the C5d domain of C5 (colored dark green) translocates away from its original position. It is long known that newly formed C5b is hydrophobic and therefore unstable in solution (DiScipio *et al*, 1983); the rapid association with C6 (within ~2.5 min; Cooper & Müller-Eberhard, 1970; Shin *et al*, 1971) is needed to form stable C5b6 complexes. We wonder whether local cleavage of C5 would allow these hydrophobic sites of C5b to directly bind to membranes (Al Salihi *et al*, 1988). Alternatively, we speculate that locally assembled C5b may have more flexibility to properly guide the localized insertion of following MAC components into the bacterial membrane. The position of C5d in Fig 9A is based on structures of preassembled C5b6 (Aleshin *et al*, 2012; Hadders *et al*, 2012). However, the exact position of C5d in locally assembled C5b may be different. Although the nature of the conformational change from C5 to C5b is similar to that observed in conversion of the highly homologous C3 into C3b (Janssen *et al*, 2006; Aleshin *et al*, 2012), C5d translocation is less pronounced than C3d (Fig 9B). Since the C5d domain is bound to a flexible arm, we speculate that the functional differences between locally assembled C5b6 and preassembled C5b6 could be due to more conformational flexibility of C5b during local assembly (Fig 9C). These hypotheses seem in contrast with a recent manuscript by Menny *et al* (2018) showing that the core of C5b remains largely unchanged during MAC assembly in liposomal membranes. However, since Menny *et al* used preassembled C5b6 to generate MAC pores, the C5b6 structure within these pores may differ from that of convertase-generated C5b6 on a bacterial surface. Potential structural differences in locally assembled C5b6 may be crucial to stably insert pores into bacterial (outer) membranes that have a very heterogenous lipid composition and varying lengths of lipopolysaccharide (LPS; Silhavy *et al*, 2010). Interestingly, Menny *et al* also describe how C6 undergoes marked domain rearrangements upon integration into the MAC. In contrast

to its conformation in preassembled C5b6 in solution, C6 in membrane-inserted MAC pores has transmembrane hairpin regions inserted into the membrane. The authors suggest that binding of C7 is needed to induce this structural change in C6 (Menny *et al*, 2018). Potentially, such structural changes in C6 occur less efficient on bacterial membranes, but are enhanced upon local C5b6 assembly, immediate binding of C7 and insertion into the membrane. In addition, a direct and stable interaction between the C5 convertase and C5b6 may be needed to maintain a different conformation of C5b6 that is lost upon release of C5b6 from the surface. Although it has been postulated that C5b6 remains bound to the C5 convertase (Morgan *et al*, 2016), it is currently difficult to assess such complexes because of limited tools to study the multi-component C5 convertase enzymes that have a surface-specific conformation (Rawal & Pangburn, 2001). Please note that the structural models of (a subunit of the) C5 convertase enzyme bound to C5, C5b, and C5b6 (Fig 9A, B, CI and CII) are not experimentally proven. Furthermore, the fact that such interactions are likely very transient (washing abrogates C5b6 activity; Fig 6) will further complicate demonstrating the existence of such intermediary complexes. Finally, different forms of MAC assemblies may exist as has been observed in liposomes, such as open versus closed pores (Menny *et al*, 2018) or clustered pores (Sharp *et al*, 2016). Potentially, the local generation of MAC pores by surface-bound C5 convertases may influence the structure or clustering of pores on bacterial membranes. Nevertheless, the here-proposed models do not exclude other scenarios of C5b6 inactivation and further studies are needed to determine what causes the rapid inactivation of C5b6.

In all, our data imply that the natural assembly of MAC via surface-specific complement activation mechanisms is essential to induce bacterial killing. Our findings that (high concentrations of) preassembled C5b6 can trigger MAC-mediated perforation of mammalian cells could be relevant for future studies on complement-mediated human diseases, where the MAC attacks the body's own cells. Some studies suggest that C5b6 can be released from activating surfaces (erroneously recognized host cell or bacterium) and associate with neighboring host cells, thereby causing unwanted "bystander" lysis (Podack & Tschopp, 1984). The information that bacterial and human cells have different sensitivities for "released" C5b6 may guide the development of complement inhibitors that specifically block bystander lysis by released C5b6 without affecting the assembly of bactericidal MAC pores.

Finally, our data also shed new light on the strongly debated mechanism of bacterial killing via the MAC. The potent bacteriolytic activity of serum was recognized in 1895 by Jules Bordet who discovered complement as a potent system that allows antibodies to directly kill bacteria (Schmalstieg & Goldman, 2009). However, the exact mechanism of MAC-mediated bacterial killing has been strongly debated (Bhakdi *et al*, 1987; Dankert & Esser, 1987; Taylor, 1992; Berends *et al*, 2014; Morgan *et al*, 2017). The fact that we now have tools to form bactericidal MAC pores in a purified manner will allow mechanistic studies to unravel how bactericidal MACs damage the bacterial cell envelope. The data with step-wise MAC formation (Fig 7) rule out the idea that formation of convertases and/or C5b-9 underneath the outer membrane is required to make pores in the inner membrane. Washing experiments show that convertase-mediated MAC assembly mainly takes place on the outer membrane. Up till the formation of C5b-8, we did not observe

**A**

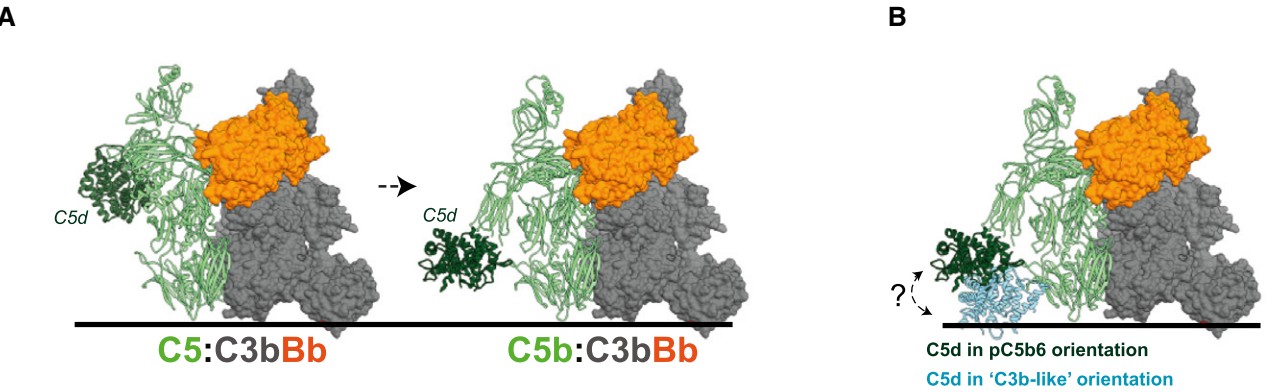

**B**

**C**    hypothetical structural models for C5b6 assembly by convertases

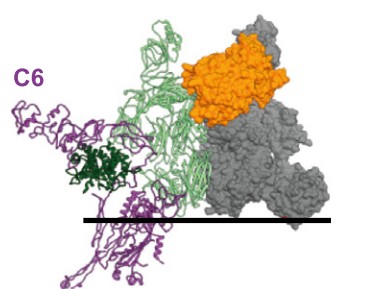
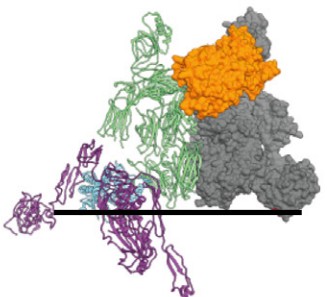
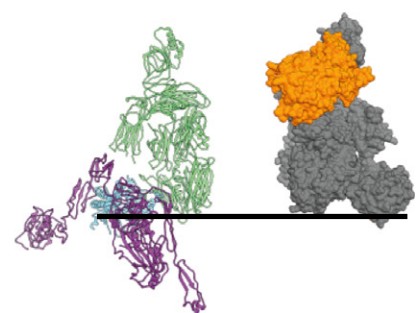

I. C5b6-conv complex
with C5d in pC5b6 orientation

II. C5b6-conv complex
with C5d in 'C3b-like' orientation

III. C5b6 released from convertase

**Figure 9. Structural model for C5b6 assembly by C5 convertases.**

A  Hypothetical model for C5 cleavage by the alternative pathway C5 convertase. The AP C5 convertase is a multimeric complex between a dimeric C3 convertase enzyme (comprised of surface-bound non-catalytic C3b in complex with protease Bb), together with additional surface-bound C3b molecules (not depicted here), which are required to strengthen the affinity for C5. Hypothetical model of C3bBb (surface representation, C3b in gray, Bb in orange) bound to substrate C5 (light green, C5d domain in dark green). C3bBb is derived from the dimeric C3bBb-SCIN complex (PDB 2WIN; Rooijakkers et al, 2009), and C5 is modeled based on superposition of the CVF-C5 complex (PDB 3PVM; Laursen et al, 2011) on the C3b molecule from C3bBb. The right panel shows C3bBb bound to C5b (light green, C5d in dark green). The structure of C5b is derived from the structure of the C5b6 complex (PDB 4A5W; Hadders et al, 2012) and superimposed on C5 from the model in the left panel.

B  Superposition of C5b with the C5d domain in the pC5b6 (dark green) and C3b-like (light blue) orientation. The C3b-like conformation of C5d was generated based on superposition of the C5d structure (extracted from the pC5b6 structure, PDB 4A5W) on the C3d domain of the second C3b subunit from the dimeric C3bBb-SCIN structure (PDB 2WIN).

C  Hypothetical structural models for C5b6 assembly by convertases. (I) Model of pC5b6 bound to C3bBb, as in (A, right). (II) Model of pC5b6, with C5d-C6 superimposed on C5d in the C3b-like orientation, as in (B). Note that this orientation allows C6 to extend further toward the membrane relative to the convertase. (III) Model in which C5b6 has dissociated from C3bBb, but adopted the orientation shown in (II). All structural models and superpositions were generated using UCSF Chimera (Pettersen et al, 2004).

extensive damage to both membranes. Addition of C9 was required to efficiently perturb the outer and inner membrane. This leaves us with two potential hypotheses for bactericidal effects at the inner membrane. First, MAC pores in the outer membrane could allow C9 to go through the pores and reach the inner membrane (as was suggested earlier; Wang et al, 2000; Dankert & Esser, 1987). Given that peptidoglycan has a pore-size distribution of 4–16 nm (Turner et al, 2013), C9 (61 kDa) might be able to reach the inner membrane where it may have cytotoxic effects (Berends et al, 2014). This would mean that pores formed in the inner membrane (by C9) are different from the pores formed in the outer membrane

(C5b-9). However, since we observed no GFP leakage from the cytoplasm of E. coli upon exposure to the MAC (Fig EV1A), it seems less likely that large pores are formed in the inner membrane. Second, inner membrane damage may directly result from MAC-dependent damage of the outer membrane. The fact that we measure influx of DNA dyes does not necessarily mean that large pores are formed in the inner membrane. Others have reported that extensive outer membrane stress by itself may cause destabilization of the inner membrane and passage of DNA dyes (Lebaron et al, 1998). Outer membrane degradation may disrupt protein connections between the outer and inner membrane that influence envelope stability

(Silhavy *et al*, 2010) or bacteria may turn on self-death machinery in response to outer membrane stress, for example via the production of regulatory RNAs (Konovalova *et al*, 2016). Exposing a bacterial knockout library to the MAC may reveal whether bacterial factors and active processes are involved in triggering OM-mediated IM damage. This hypothesis would be consistent with the idea that local assembly of the MAC is required to more efficiently perforate the outer membrane and thereby indirectly cause inner membrane damage. Since the outer membrane is a major permeability barrier for most antibiotics, the mechanisms by which complement disrupts Gram-negative cell envelopes may hold information crucial for development of antimicrobial strategies against drug-resistant Gram-negative bacteria (Laxminarayan *et al*, 2013).

# Materials and Methods

### Serum, reagents, and bacterial strains

Normal human serum and heat-inactivated serum were obtained from healthy volunteers as previously described (Berends *et al*, 2013). Sera depleted of complement factors and complement factors C5b6, C7, and C8 were obtained from Complement Technology. His-tagged C5, C6, C9, FB, and FD were expressed in and purified from HEK293E cells (U-Protein Express). Lysozyme-depleted serum was prepared as described below. $_{Peri}$mCherry$_{/cyto}$GFP *E. coli* was prepared by transforming a pPerimCh plasmid into *E. coli* MG1655. pPerimCh was modified from plasmid pFCcGi containing a constitutively expressed mCherry and a L-arabinose inducible GFP (kindly provided by Sophie Helaine). A pelB leader was added in front of the mCherry sequence to direct mCherry to the periplasmic space. Gram-negative isolate *S. maltophilia* 566954.1 was obtained from the diagnostic Medical Microbiology department of the University Medical Center Utrecht. OmCI was produced in HEK293E cells and purified as previously described (Nunn *et al*, 2005). Eculizumab was kindly provided by Frank Beurskens (Genmab, Utrecht, The Netherlands). Normal concentrations of MAC proteins in 100% human serum are: ± 375 nM C5, 550 nM C6, 600 nM C7, 350 nM C8, and 900 nM C9. Hybridoma cells producing MoAb bH6 to C3b were kindly provided by Peter Garred (University of Copenhagen) and Tom Eirik Mollnes (University of Oslo). Antibodies were purified as previously described (Garred *et al*, 1988). Purified antibodies were labeled with Alexa Fluor 488 reactive molecules (ThermoFisher) following the supplier's protocol.

### Preparation of lysozyme-depleted serum

The sequence of lysozyme inhibitor LprI of *M. tuberculosis* without signal peptide was synthesized (IDT; Sethi *et al*, 2016). The sequence was cloned into a modified, N-terminal His-tag followed by a TEV cleavage site, pRSETB vector (Thermofisher) digested with BamHI/NotI using Gibson assembly (NEB). The protein of interest was expressed in BL21(DE3; Thermofisher) by adding 1 mM IPTG. LprI was isolated under denaturing conditions using a Histrap column (GE Healthcare). For lysozyme depletion, a Histrap column was charged with $CoCl_2$ and loaded with 1.1 mg His-tagged LprI. The protein was covalently linked to the column by adding 0.05% $H_2O_2$ in PBS for 1 h at RT. After washing with PBS, 4 ml of human pooled serum in the presence of 10 mM EDTA was loaded on the column. To the flow-through, containing lysozyme-depleted serum, 10 mM $CaCl_2$ and 10 mM $MgCl_2$ were added. Lysozyme-depleted serum was analyzed for complement activity by performing a CH50 (Fig EV1C). Complement activity was comparable to serum before lysozyme depletion. Successful depletion was demonstrated using a lysozyme ELISA (Abcam), which showed over 99% depletion of lysozyme.

### Hemolytic and liposome assay

Rabbit erythrocytes ($1 \times 10^8$/ml) or liposomes (Wako CH50 Auto kit) were incubated with buffer or purified MAC components for 30 min, non-shaking at 37°C. 10 nM C5b6, 20 nM C7, 20 nM C8, and 100 nM C9 were used. Incubations were done in Veronal + 2.5 mM $MgCl_2$ and 0.5 mM $CaCl_2$, VBS++. As positive controls, erythrocytes were incubated with MQ, liposomes with 0.05% PBS-Tween. After incubations, erythrocytes were spun down and absorbance of the supernatant at OD405 nm was measured. The percentage of lysed rabbit erythrocytes was calculated by comparing the OD405 nm of the test sample with the OD405 nm of the Milli-Q control sample, which was set at 100% lysis. NADH production as a result of G6PDH leakage from liposomes was determined by measuring the absorbance at OD340 nm. Human erythrocytes were collected from freshly drawn blood, which was spun down, washed three times in PBS after which the cells were collected. Cells ($1 \times 10^8$/ml) were exposed to a concentration range of C5b6 or C5 and C6 in the presence of 100 nM C7. After washing, cells were exposed to 20 nM C8 and 100 nM C9 for 30 min and spun down, and absorbance of the supernatant at OD405 nm was measured.

### Convertase labeling of bacteria

In all experiments, bacteria were grown overnight (o/n) in Lysogeny broth (LB) medium (containing 50 μg/ml ampicillin for $_{Peri}$mCherry$_{/cyto}$GFP *E. coli* MG1655). Next day, subcultures were grown to mid-log phase (OD$_{660}$~0.5), washed and resuspended in RPMI + 0.05% HSA. Unless stated differently, all incubations with bacteria were performed in RPMI + 0.05% HSA. Bacteria with OD$_{660}$~0.1 were incubated with 10% C5 depleted serum (ΔC5 serum) for 30 min at 37°C, washed and resuspended to OD$_{660}$~0.05. Complement activation in ΔC5 serum was blocked by heat inactivation or by adding 5 μM compstatin. For labeling with classical pathway convertases, bacteria were incubated with ΔFB serum in the presence of 20 μg/ml OmCI for 30 min at 37°C. C5a generation was measured in a calcium mobilization assay as previously described (Bestebroer *et al*, 2010).

### HAP1 cell lysis assay

ΔCD46/ΔCD55/ΔCD59 HAP1 cells (Thielen *et al*, 2018) were kindly provided by Sanquin (Amsterdam, the Netherlands). Cells were cultured in Iscove's modified Dulbecco's medium (IMDM) supplemented with 10% fetal bovine serum, penicillin, and streptomycin (Gibco) at 37°C with 5% $CO_2$. Cells were washed in PBS and harvested using trypsin. The collected cells were washed 2 times in PBS and resuspended to a concentration of $4 \times 10^6$ cells/ml in VBS++. These cells were incubated with buffer (VBS++) or 25% C5

depleted serum for 30 min at 37°C under shaking conditions. Unlabeled or opsonized cells were washed twice in VBS++ (2 min at 300 g) after which 50 µl (100,000 cells/well) was incubated with 100 nM C5b6 or C5 and C6 in the presence of 100 nM C7, 10 µg/ml FB, and 1 µg/ml FD for 15 min at 37°C. Cells were washed twice and incubated with 100 nM C8 and C9 for 30 min at 37°C in the presence of Sytox green. Cells were diluted 1:1 with PBS and Sytox intensity was measured by flow cytometry.

### Membrane permeabilization and bacterial viability assay

Unlabeled or convertase-labeled bacteria were prepared as described above and incubated with 10% serum or purified MAC components. Unless stated differently, 10 nM C5, 10 nM C6, 10 nM C5b6, 20 nM C7, 20 nM C8, and 100 nM C9 were used. These concentrations are similar to protein concentrations in 2–10% serum. All incubations with serum and purified MAC components were done for 30 min at 37°C. However, in experiments with preassembled C5b6, a washing step was introduced to prevent fluid phase MAC formation. Incubations with preassembled C5b6 were done in the presence of C7 for 15 min at 37°C. Samples were washed 3 times after which C8 and C9 were added for 30 min at 37°C. The washing step after C7 was also introduced when uncleaved C5 and C6 were tested in the same experiment as preassembled C5b6. For the C5b6 stability experiments, convertase-labeled bacteria were incubated with C5 and C6 in the presence or absence of C7 for 15 min at 37°C. Then, bacteria were washed or incubated with RPMI or 10 µg/ml Eculizumab for another 15 min at 37°C. Subsequently, the remaining MAC components were added for 30 min at 37°C, after which samples were diluted and measured by flow cytometry. In all experiments, 2.5 µM Sytox Blue Dead Cell Stain (Thermofisher) was added to the final incubation step of the experiments. mCherry, Sytox, and GFP intensities were measured by a MACSQuant flow cytometer.

### Bacterial viability assay

After incubating bacteria with serum or purified MAC components as described above, samples were serially diluted in PBS and plated onto LB agar plates. Colonies were counted after overnight incubation.

### Complement deposition and trypsin treatment

Convertase-labeled bacteria were prepared as described above. To measure C3b deposition, bacteria were incubated with 3 µg/ml Alexa-488 labeled mouse-anti-C3b (described above) for 30 min at 4°C. C9 deposition was measured by C-terminal sortagging C9-LPETG-His with GGG-N$_3$, which was then coupled to Cy3-DBCO or Cy5-DBCO. For trypsinization, bacteria were first incubated with MAC components for 30 min and subsequently treated with 20 µg/ml trypsin for 15 min at 37°C. C3b and C9 deposition was measured by flow cytometry.

### Confocal microscopy

Samples were prepared as described above, concentrated to OD$_{600}$~1.5, and dried onto 1% agar pads. To-pro-3 (1 µM,

Thermofisher) was used as a DNA dye. Agar pads were placed onto a coverslip and samples were imaged using a Leica SP5 confocal microscope with a HCX PL APO CS 63×/1.40–0.60 OIL objective (Leica Microsystems, the Netherlands).

### Structured illumination microscopy

8-well microslide (Ibidi) chambers were washed three times with 500 µl 1 M HCl/70% EtOH solution and rinsed with 500 µl MQ for three times. Chambers were coated with 150 µl 1 M sodium acetate/0.01 M NaOH and 4 µl Cell-Tak solution (Corning) for 20 min (RT), washed three times with MQ and dried. $_{Peri}$mCherry/$_{cyto}$GFP E. coli were grown to mid-log phase in the presence of 0.1% arabinose, washed three times in MQ, and immobilized onto the coverslip for 30 min. Samples were washed with RPMI + 0.05% HSA, and images were obtained using the GE Healthcare Life-Sciences "Deltavision OMXV4 blaze" microscope using 60× Olympus lens (U-PLAN APO, NA 1.42) and immersion oil 1.516 (Cargille laboratories). GFP and mCherry signals were measured using the 488-nm and 561-nm lasers, respectively, with suited dichroics and emission filter setting of 528/48 and 609/37. Reconstructions and registrations were performed using softWoRx (GE healthcare).

### Atomic force microscopy

Mid-log phase bacteria (E. coli—MG1655/BL21) were washed three times in PBS or PB (10 mM), concentrated four times, and immobilized onto Cell-Tak or poly-L-lysine (0.01%) covered glass slides (Corning/Sigma-Aldrich). Both methods of immobilization were found to yield equivalent results. Care was taken not to allow the bacteria to dry out during immobilization. Immobilized bacteria were incubated with 10% ΔC5 serum in VBS++ containing 0.1% bovine serum albumin (BSA) for 20 min at 37°C. Glass slides with immobilized, serum-treated bacteria were rinsed 3 times with PBS/PB. Bacteria were then treated with a solution of 25 µg/ml C5b6, 20 µg/ml C5, 12 µg/ml C6, 12 µg/ml C7, 50 µg/ml FB, and 5 µg/ml FD (all in VBS++) and incubated for 5 min at RT. Following this, 15 µg/ml C8 and 70 µg/ml C9 in VBS++ were added for a 10–40 min incubation at 37°C. Glass slides with immobilized, treated bacteria were rinsed 3 times in PBS/PB before atomic force microscopy imaging.

Atomic force microscopy topographic images of E. coli (MG1655) in Figs 8D and EV5 were obtained using a Nanowizard III AFM with an UltraSpeed head (JPK, Germany) operated in liquid at room temperature. The microscope was operated in intermittent contact mode using FastScan-D probes ($k$ = 0.25 N/m; Bruker). Images were processed using Gwyddion (Nečas & Klapetek, 2012) (http://gwyddion.net/) for 1$^{st}$ order line-by-line flattening to remove tilt. Images of bacterial surfaces were then processed using an additional 2$^{nd}$ order polynomial fit to remove the curvature of the bacteria. The data on E. coli (BL21) shown in Fig 8C were obtained using a Bruker FastScan Bio AFM, operated in PeakForce Tapping mode in liquid at 8 kHz using FastScan-D probes ($k$ = 0.25 N/m; Bruker). Images were processed using NanoScope Analysis (Bruker) for 1$^{st}$ order line-by-line flattening to remove tilt. Images of E. coli surfaces were processed using an additional 2$^{nd}$ order polynomial fit to remove the curvature of the bacteria and using a ~1.5 nm low-pass filter to remove high frequency noise. Images are displayed as height data in 3D. Cross-sectional analysis was performed in

NanoScope Analysis and plotted in Origin (OriginLab), along the dotted lines indicated in Fig 8C. For measurements of bacterial height, the cross-sectional profile was taken over a 500 nm width to obtain an average for the bacterium.

### Data analysis and statistical testing

Flow cytometry data were analyzed in FlowJo, and percentage positive cells were based on gating for positive controls. Graphpad 6.0 was used for graph design and statistical analysis. Statistical analysis was done using a ratio paired two-tailed *t*-test or a one-way ANOVA as indicated in the figure legends, in which each condition was compared with a control sample (buffer treated) unless stated differently. Three experimental replicates were performed to allow statistical analysis.

**Expanded View** for this article is available online.

### Acknowledgements
We would like to acknowledge Piet Gros, Paul Parren, and Jos van Strijp for proofreading the manuscript and Zvi Fishelshon for fruitful discussions; Jannik Luebke and Richard Thorogate for assistance with AFM experiments; Richard Wubbolts for assistance with microscopy, U-Protein Express BV for protein expression facilities. We thank Ilse Jongerius and Diane Wouters (Sanquin Research, Department of Immunopathology, Amsterdam, the Netherlands, and Landsteiner Laboratory, Amsterdam UMC, University of Amsterdam, Amsterdam, the Netherlands) for providing ΔCD46/ΔCD55/ΔCD59 HAP1 cells; John Lambris (University of Philadelphia) for providing compstatin and Frank Beurskens (Genmab, Utrecht) for providing Eculizumab. The work was funded by: an ERC Starting grant (639209-ComBact, to S.H.M.R); the EMBO Young Investigator Program (3418 to S.H.M.R); UK RCUK | Biotechnology and Biological Sciences Research Council (BBSRC) and MRC project grants (BB/N015487/1 and MR/R000328/1, to B.W.H.); UK RCUK | Engineering and Physical Sciences Research Council (EPSRC) and MRC fellowships (EP/M507970/1 to E.S.P.; EP/M506448/1 and MR/R024871/1 to A.L.B.P.); UCL Impact Award (EP/M506448/1 to I. B.); and UK EPSRC investments in AFM equipment (EP/K031953/1 via the IRC in Early-Warning Sensing Systems for Infectious Diseases; and EP/M028100/1).

### Author contributions
DACH, BWB, ESP, IB, DJD, ETMB, ALBP, BWH, and SHMR designed research; DACH, BWB, ESP, IB, MR, DJD, ETMB, and ALBP performed experiments; BWB developed flow cytometry methodologies for outer/inner membrane permeabilization; RDG generated atomic models of C5 convertases and C5b6; DACH, ESP, IB, and ALBP performed AFM experiments; DACH, BWB, ESP, IB, DJD, ALBP, BWH, and SHMR analyzed the data; and DACH, BWB, ESP, IB, ALBP, BWH, and SHMR wrote the paper. ALBP, BWH, and SHMR supervised the project. All authors approved the final version of the manuscript.

### Conflict of interest
The authors declare that they have no conflict of interest.

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
