## [Review Process File · The EMBO Journal]

Bacterial killing by complement requires membrane attack complex formation via surface-bound C5 convertases

Dani A. C. Heesterbeek, Bart W. Bardoel, Edward S. Parsons, Isabel Bennett, Maartje Ruyken, Dennis J. Doorduyn, Ronald D. Gorham Jr., Evelien T.M. Berends, Alice L. B. Pyne, Bart W. Hoogenboom and Suzan H. M. Rooijackers.

Review timeline:

Submission date:	18 th May 2018
Editorial Decision:	30 th July 2018
Revision received:	23 rd October 2018
Editorial Decision:	28 th November 2018
Revision received:	5 th December 2018
Accepted:	14 th December 2018

Editor: Elisabetta Argenzio

Transaction Report:

1st Editorial Decision

30th July 2018

Thank you for submitting your manuscript on a role for C5 convertases during membrane attack complex formation and bacteria killing to The EMBO Journal. My apologies for the extended duration of the review process for this manuscript. Three referees were originally assigned to your manuscript, however one of them did not return his/her report even after repeated messages and a fourth referee had to be involved. We have now received three referee reports on your study and these are enclosed below for your information.

As you can see, while the referees consider the findings novel and interesting, they also raise critical points that need to be addressed before they can support publication at The EMBO Journal. In particular, referee #1 is concerned that the study fails to address the differences between the structure and function of the two C5 isoforms and the role of C5 convertases in human cells. Referee #2 stresses that the contribution from proteins other than C5 convertases, and the mechanism of pore formation on the outer membrane and inner membrane destabilization need to be further investigated. Finally, referee #3 requests you to test the equal distribution of C5 convertases on bacteria and the existence of intermediate complexes between C5 isoforms and downstream MAC components.

Addressing these issues through decisive additional data as suggested by the referees would be essential to warrant publication in The EMBO Journal. Given the overall interest of your study, I would thus like to invite you to revise the manuscript in response to the referee reports.

REFeree REPORTS

Referee #1:

The work of Heesterbeek et al asks the pertinent question how the membrane attack complex (MAC) of the complement system kills bacteria. They show that a membrane-bound C5 convertase is necessary for killing. Pre-formed C5b can generate MAC creating a pore in the outer membrane but inefficient to lyse the inner membrane of the bacteria. Although the experimental system used is elegant, the authors only observe the discrepancy of the action of locally-formed versus distantly-generated C5 but do not explain it. At the current stage the manuscript is descriptive and in its current form does not bring significant novel information. The results are interesting, but for a restricted number of specialists.

In the manuscript it is unclear what the differences between the structure and function of the two forms of C5 are. Why only locally formed C5b can generate bacteria-killing MAC?

It is suggested that the findings may guide the development of therapeutic approaches to improve bacterial killing by targeting MAC formation at the membrane. Nevertheless, it is unclear to this reviewer how the knowledge that MAC has to be formed on the membrane (which is intuitive and given in all textbook schemas), will help in this process. There are no experiments to illustrate what could be done differently in terms of monoclonal antibodies targeting for example (as suggested by the authors) in order to improve the bactericidal activity.

The rapid loss of cell-killing properties of MAC upon release from the surface could be a regulatory mechanism, avoiding bystander injury of host cells, when complement is fully activated at bacterial surface. What is the situation with the human cells? Do they sense the difference between C5b formed at their surface or in the vicinity?

Referee #3:

The manuscript by Heesterbeek et al. focuses on investigating the mechanism by which MAC perturbs bacterial cell envelope. Even though MAC is well-known to have bactericidal activity, the underlying mechanism is still not fully understood. Using a variety of complementary approaches, the authors found that C5 conversion (thus MAC formation) on the surface by C5 convertase is required for pores formation and killing of bacteria. Overall, the manuscript is well written and the conclusions are properly supported by the experimental results. The findings in this manuscript are quite novel and further improves our understanding of how the complement system works.

Major comments:

Fig 3b: To rule out the potential contribution from proteins other than C5 convertase, is it possible to perform the labeling with convertase negative serum and test for MAC activity?

Fig 3d indicated that C5-C8 complex showed some bactericidal activity in the absence of C9. Yet in Fig 4d, the same C5-C8 complex did not cause any significant leakage of both membranes. Can the authors comment on this observation?

Fig 5: The result in Fig 5b indicated that the C5b6 MAC can still form on the surface, albeit not properly inserted as shown by the trypsin treatment. However, there were almost no C5b6 MAC in the AFM image (Fig 5e). Can the authors explain this discrepancy?

Line 220: "...a higher efficiency of MAC perforation of the outer membrane, thereby destabilizing the inner membrane". The authors need to clarify how pore formation on the outer membrane can lead to inner membrane destabilization.

Minor comments:

SFig. 2b: The authors never mentioned the concentration of complement proteins in blood/serum. This information will be useful to help the readers compare experiments with serum and purified proteins (for example: Fig 2d).

Line 174: "...our results imply that C5b6 loses bactericidal capacity upon release from the surface". The authors never tested the activity of C5b6 upon release from the surface directly. The data indicated that C5b6 was not bactericidal when produced in solution.

Referee #4:

Overall

The paper addresses the critical question of how insertion of MAC in the outer membrane can lead to inner membrane permeabilization, which is a major unresolved question regarding the complement terminal pathway. The manuscript convincingly demonstrates that MAC assembly through *in situ* C5 cleavage and downstream MAC assembly is critical for inner membrane permeabilization and subsequent bactericidal activity. Elegant experimental systems are described that allows the authors to follow outer and inner membrane permeabilization with flow cytometry and confocal microscopy. A unique feature of the manuscript is also the beautiful AFM pictures of the MAC, this a real strength and a significant advance in the complement field that may become a valuable supplement to present and future studies of the terminal pathway with cryo-EM and cryo tomography. The work conducted is technically excellent and the figures+legends are in general very good and easy to follow

Overall, the topic and the results are of strong interest for investigators of the fundamental mechanism of the terminal pathway in the complement system, and as such it is of fundamental interest to investigators in the field of complement biology, innate immunity and microbiology. It is also likely to be well received amongst investigators and the multiple companies pursuing therapeutic modulation of the complement system since the terminal pathway is a very well validated target. This being said, the experimental evidence should be strengthened in a revised manuscript and the discussion reworked and extended.

Major points

Are the C5b6 MACs and C5 convertase MACs distributed identically on the bacteria? This may be relevant for the difference in bactericidal activity. The authors has addressed previously the distribution of MACs on bacteria and shown that it non-random

Another major question is whether the requirement for the C5 convertase is due to a stable and direct complex formed between the C5 convertase and C5b6 and downstream MAC intermediates. This is not shown directly. Eculizumab or its Fab appears to be an obvious reagent to settle this question

Then, if such an intermediate is shown to exist

1) at which step of MAC assembly can the C5 convertase become dissociated from the MAC intermediate without compromising the ability of MAC to lyse the inner membrane ? eculizumab could be added after addition of C7 or C8 to dissociate the C5 convertase from C5b in these MAC intermediates and then afterwards the remaining components C8+C9 or C9 could be added to complete the MAC.

2) what is the temporal stability of such a complex (if existing)

Also the authors has just published an elegant study where they show that selective inhibition of the C5 convertase with complement evasion proteins and complement regulators is possible. Inclusion of these proteins in the same directions as suggested above with eculizumab may also be helpful for understanding better WHY and WHEN the C5 convertase is critical.

Both points above are mentioned in the discussion as future directions, but with the methodology so well established it should be possible to add this information to the revised manuscript in a reasonable time frame

There are now very good atomic models of the MAC and the C5 convertase may to some degree be approximated by the existing model of the C3 convertase. Hence, it is possible to construct a reasonable model of C5b6 and MAC bound to the convertase. This is highly relevant for the paper as this will suggest how the convertase orients C5b6 and downstream MAC intermediates.

Interpretation or at least a discussion of the experimental data taking into account all the known structural data concerning C5b6, MAC and the C5 convertase would strengthen the manuscript. The discussion is very brief. In addition to the inclusion of structural knowledge, a more elaborate discussion of the possible mechanism of inner membrane permeabilization is needed and also experimental approaches that may be used to dissect this mechanism in future work should be added. In the discussion, it is also unclear why MAC mediated killing of cancer cells is mentioned, here only a single membrane is present and there is no experiments with mammalian cells in the manuscript.

Minor points

Line 67, ref 2 is not the most relevant

Line 73-75. That MAC form toroid shapes in the membrane is old knowledge, not recent. Either reformulate to emphasize the cryoEM findings that give very detailed information or insert older references for the overall toroid shapes

Line 108:112. Why does the absence of lysozym increase influx of Sytox?

Lines 167:168. Emphasize the difference between permeation and permeabilization better, otherwise the conclusion may appear unjustified

Line 181:182. Provide a more direct comparison than just "similar", this is quite important data.

Perhaps panels 5b-c could be combined?

Line 183:184. It appears from the methods that trypsin was added after MAC formation, that should be mentioned

Line 200-: AFM, briefly explain the difference between the "height" and the "phase" panels images to the non-expert

Suppl fig 6. Label the panels with the immobilization protocol used in each case

1st Revision - authors' response

23rd October 2018

POINT-TO-POINT REPLY

Referee #1:

Point 1: *“In the manuscript it is unclear what the differences between the structure and function of the two forms of C5 are. Why only locally formed C5b can generate bacteria-killing MAC?”*

Answer 1: The revised manuscript contains new experimental data explaining why local assembly of C5b(6) is essential for bacterial killing. Our data indicate that once formed, C5b6 rapidly loses capacity to form bactericidal pores; therefore, bacterial killing requires both *in situ* conversion of C5 and immediate insertion of C5b67 into the membrane (**new Fig. 6**). Thus, in addition to our conclusions in the previous submission, we now know that only locally formed C5b can generate bactericidal MACs, because once generated, C5b(6) only remains fully functional over a limited time window and needs to rapidly bind C7 onto the bacterial membrane. At present, the available structural data is insufficiently detailed to fully clarify differences in structure of the two forms of C5(b), but enough to propose plausible structural models that elaborate on potential structural differences between locally assembled C5b6 and preassembled C5b6 (**new Fig. 9**). In short, we propose that newly formed C5b by a surface-bound convertase either has 1) transient activity to associate to membranes, or 2) has more flexibility to properly guide the insertion of C5b6 to the bacterial membrane (**new Fig. 9** and an elaborate discussion section on page 10&11).

Point 2: *“It is suggested that the findings may guide the development of therapeutic approaches to improve bacterial killing by targeting MAC formation at the membrane. Nevertheless, it is unclear to this reviewer how the knowledge that MAC has to be formed on the membrane (which is intuitive and given in all textbook schemas), will help in this process. There are no experiments to illustrate what could be done differently in terms of monoclonal antibodies targeting for example (as suggested by the authors) in order to improve the bactericidal activity.”*

Answer 2: We have removed this statement from the abstract since the referee is right that our results do not directly result in better anti-bacterial therapies. We have now replaced the sentence with: *“These studies provide basic molecular insights into MAC assembly and bacterial killing by the immune system.”* Although it indeed seems intuitive that MAC has to be formed on the cell membrane, our findings that C5b6 rapidly loses bactericidal capacity was previously unrecognized. Also, the information that bacterial and human cells have different sensitivities for ‘released’ C5b6 could be relevant to better understand mechanisms of complement-related disorders.

Point 3: *“The rapid loss of cell-killing properties of MAC upon release from the surface could be a regulatory mechanism, avoiding bystander injury of host cells, when complement is fully activated at bacterial surface. What is the situation with the human cells? Do they sense the difference between C5b formed at their surface or in the vicinity?”*

Answer 3: We thank the referee for raising this interesting point. We now included data to study whether human cells ‘sense’ the difference between C5b6 formed at their surface or in the vicinity. MAC was assembled on convertase-labeled human cells via uncleaved C5 and C6 or preassembled C5b6 (**new SFig. 4**). Interestingly, we observed that local assembly of C5b6 was not essential to induce pore formation in human cells. This is fully consistent with the lytic function of C_{5b6} MAC in liposomes (**SFig. 2a**) and mammalian erythrocytes (**new Fig. 2b**) and (**SFig. 2b**). Given that C_{5b6} MAC (Fig. 2d) and Conv- C_{5b6} MAC (**new Fig. 5d**) pores also perturb the bacterial outer membrane, but not the inner membrane, these results strongly suggest that local C5b6 assembly is particularly important to generate pores that damage the complete, composite cell envelope of Gram-negative bacteria. These experiments highlight interesting differences in pore formation between human and bacterial membranes. As suggested by the referee, the fact that purified MAC pores can lyse human cells could suggest that released C5b6 from a bacterium may cause bystander injury of host cells. Since we think this information could be useful for future understanding and therapeutic prevention of unwanted bystander lysis of host cells, we include this in the discussion.

Referee #2:

Point 1: *“Fig 3b: To rule out the potential contribution from proteins other than C5 convertase, is it possible to perform the labeling with convertase negative serum and test for MAC activity?”*

Answer 1: We are grateful to the referee for pointing this out. The original manuscript included a control with convertase-negative serum (heat-inactivated), but this was not properly labeled in the graph (this is now corrected). Furthermore, we have now included another strategy to generate convertase negative serum (specific inhibition by compstatin) and we verify that the two approaches to make convertase negative serum were indeed successful (new **SFig. 3c** and new **SFig. 3d**). Both the killing (old **Fig. 3c** and **3d**) and Sytox influx experiments (new **Fig 4b**) show that there is no killing/inner membrane damage in convertase-negative serum, thus ruling out the potential contribution from proteins other than the C5 convertases.

Point 2: *“Fig 3d indicated that C5-C8 complex showed some bactericidal activity in the absence of C9. Yet in Fig 4d, the same C5-C8 complex did not cause any significant leakage of both membranes. Can the authors comment on this observation?”*

Answer 2: This can be explained by the different timing of these two experiments. Influx of sytox is measured after 1 hour; bacterial viability was determined after overnight growth of bacteria on agar plates.

Point 3: *“Fig 5: The result in Fig 5b indicated that the C5b6 MAC can still form on the surface, albeit not properly inserted as shown by the trypsin treatment. However, there were almost no C5b6 MAC in the AFM image (Fig 5e). Can the authors explain this discrepancy?”*

Answer 3: We think that the C_{5b6} MAC pores are not well inserted into the membrane and therefore difficult to resolve by AFM. We now better clarify this in the results section on page 9&10, line 279-282: *“This is consistent with previous AFM experiments invasiveness of the AFM measurement and/or insufficient temporal resolution”*

Point 4: *Line 220: “...a higher efficiency of MAC perforation of the outer membrane, thereby destabilizing the inner membrane”. The authors need to clarify how pore formation on the outer membrane can lead to inner membrane destabilization.*

Answer 4: The revised manuscript now includes new data showing that local assembly of MAC indeed results in more efficient perforation of the outer membrane (new **SFig. 6a**). Furthermore, we have extended our discussion section on page 12 to explain how pore formation on the outer membrane can lead to inner membrane destabilization. Please see: *“This leaves us with two potential hypotheses (page 12, line 370) till “.....are involved in triggering OM-mediated IM damage’ (page 13, line 387).*

Minor comment 1: “SFig. 2b: The authors never mentioned the concentration of complement proteins in blood/serum. This information will be useful to help the readers compare experiments with serum and purified proteins (for example: Fig 2d).”

Answer: Both the methods section and legend of figure 2 now state the concentrations of MAC proteins in 100% normal serum: Normal concentrations of MAC proteins in 100% human serum are: \pm 375 nM C5, 550 nM C6, 600 nM C7, 350 nM C8 and 900 nM C9. (Methods, page 13, line 407-408; legends, page 23, line 728)

Minor comment 2: “Line 174: “...our results imply that C5b6 loses bactericidal capacity upon release from the surface”. The authors never tested the activity of C5b6 upon release from the surface directly. The data indicated that C5b6 was not bactericidal when produced in solution.”

Answer: Although we agree with the referee that we didn’t test C5b6 released from the bacterial surface, preassembled C5b6 does resemble ‘released’ C5b6 since it was generated by activating C5 and C6 in human serum in the absence of C7 on zymosan surfaces and subsequently purifying ‘released’ C5b6 complexes from the supernatant. We apologize that this was not clear and now better explain how preassembled C5b6 was produced in Results (page 5, line 118-119) and Discussion (page 10&11, line 312-314).

Referee # 4:

Point 1: Are the C5b6 MACs and C5 convertase MACs distributed identically on the bacteria? This may be relevant for the difference in bactericidal activity. The authors has addressed previously the distribution of MACs on bacteria and shown that it non-random

Answer 1: The revised manuscript now includes confocal microscopy data showing no differences in the distribution of C_{5b6}MACs and conv-MACs on bacteria. We have now included these data as a new figure (**Fig. 8b**).

Point 2: Another major question is whether the requirement for the C5 convertase is due to a stable and direct complex formed between the C5 convertase and C5b6 and downstream MAC intermediates. This is not shown directly. Eculizumab or its Fab appears to be an obvious reagent to settle this question. Then, if such an intermediate is shown to exist

1) at which step of MAC assembly can the C5 convertase become dissociated from the MAC intermediate without compromising the ability of MAC to lyse the inner membrane? eculizumab could be added after addition of C7 or C8 to dissociate the C5 convertase from C5b in these MAC intermediates and then afterwards the remaining components C8+C9 or C9 could be added to complete the MAC.

2) what is the temporal stability of such a complex (if existing)

Also the authors has just published an elegant study where they show that selective inhibition of the C5 convertase with complement evasion proteins and complement regulators is possible. Inclusion of these proteins in the same directions as suggested above with eculizumab may also be helpful for understanding better WHY and WHEN the C5 convertase is critical.

Answer 2: We have performed the suggested experiments to investigate whether complex formation between convertases and C5b6 is required for bacterial killing.

First, we used the C5 inhibitor Eculizumab to address this question. Convertase-labeled bacteria were incubated with components C5, C6, C7, C8 and C9. When Eculizumab was added to the incubation mixture between C5/6 and C7/8/9, we observed a strong inhibition of bactericidal MAC formation by Eculizumab (**new Fig. 6**). We think that the inhibition by Eculizumab in this experiment can be interpreted in two ways.

- 1) Eculizumab could dissociate C5b6 from the convertase
- 2) Eculizumab could prevent formation of new C5b6 molecules (since it blocks binding and cleavage of C5 by the convertase).

Thus, this experiment does not proof existence of convertase-C5b6 complexes, since it cannot be determined whether Eculizumab stops C5 conversion OR breaks up potential convertase-C5b6 complexes. Also other inhibitors will not settle this question since all known convertase and C5 inhibitors will block both processes.

Second, we repeated the experiment but, instead of using Eculizumab, we introduced a washing step following C5b6 assembly (to rule out generation of new C5b6). Interestingly, washing of bacteria

also prevented bactericidal MAC formation (**new Fig. 6**). Thus, if there is a direct interaction between the convertase and C5b6, such complexes are extremely labile.

Interestingly, when C7 was present during C5b6 assembly, we observed that both Eculizumab and a washing step did not abrogate bactericidal MAC formation (**new Fig. 6**). This suggests that the C5b6 complex efficiently induces bacterial killing only when it can immediately proceed with C7 assembly at the bacterial surface. These new data imply that local assembly of C5b6 is needed to generate a transiently active form of C5b6 that should immediately react with the membrane. Otherwise, C5b6 will rapidly lose the capacity to form bactericidal pores. Altogether we think that these experiments yield critical insights into understanding why the convertase is essential.

Point 3: *There are now very good atomic models of the MAC and the C5 convertase may to some degree be approximated by the existing model of the C3 convertase. Hence, it is possible to construct a reasonable model of C5b6 and MAC bound to the convertase. This is highly relevant for the paper as this will suggest how the convertase orients C5b6 and downstream MAC intermediates. Interpretation or at least a discussion of the experimental data taking into account all the known structural data concerning C5b6, MAC and the C5 convertase would strengthen the manuscript.*

Answer 3: As suggested by the referee, we have now included structural models demonstrating how C5 convertases cleave C5 into C5b(6) (**new Fig 9**). Using these models, we postulate on potential structural differences between locally assembled C5b6 and preassembled C5b6. In short, we propose that newly C5b formed by a surface-bound convertase either has 1) transient activity to associate to membranes, or 2) has more flexibility to properly guide the insertion of C5b6 to the bacterial membrane. We did not include structural models of convertase bound to MAC since we have no evidence that this interaction (if existing) remains up to formation of MAC assembly.

Point 4: *The discussion is very brief. In addition to the inclusion of structural knowledge, a more elaborate discussion of the possible mechanism of inner membrane permeabilization is needed and also experimental approaches that may be used to dissect this mechanism in future work should be added. In the discussion, it is also unclear why MAC mediated killing of cancer cells is mentioned, here only a single membrane is present and there is no experiments with mammalian cells in the manuscript.*

Answer 4: The discussion is now significantly longer (from 1 page to 2,5 pages) and includes suggested structural models, discussion of possible mechanisms of inner membrane damage and suggestions to dissect this mechanism. Furthermore, the statement about cancer cells has been removed.

Minor comment 1: *Line 67, ref 2 is not the most relevant*

Answer: We replaced this reference with a more general review on classical pathway activation (Kang et al, Adv. Exp.Med. Biol. 653, 117–128 (2009)).

Minor comment 2: *Line 73-75. That MAC form toroid shapes in the membrane is old knowledge, not recent. Either reformulate to emphasize the cryoEM findings that give very detailed information or insert older references for the overall toroid shapes.*

Answer: This sentence has been reformulated as follows: “Recent *in-vitro* structural studies^{14,15} have revealed *detailed information on how* MAC proteins form toroid-shaped pores”. (page 3, line 74)

Minor comment 3: *Line 108:112. Why does the absence of lysozym increase influx of Sytox?*

Answer: This probably has a technical reason; lysozyme-depleted has slightly higher complement activity (as shown in **SFig. 1c**), probably caused by the difficult to control restoration of $\text{Ca}^{2+}/\text{Mg}^{2+}$ after EDTA treatment (needed to maintain complement activity during lysozyme affinity depletion).

Minor comment 4: *Lines 167:168. Emphasize the difference between permeation and permeabilization better, otherwise the conclusion may appear unjustified*

Answer: This sentence has been rephrased as follows: “This suggests that assembly of bactericidal MAC pores not requiring the stepwise assembly of new convertases or C5b-9 pores.” (Page 8, line 240-243).

Minor comment 5: Line 181:182. Provide a more direct comparison than just "similar", this is quite important data. Perhaps panels 5b-c could be combined?

Answer: As suggested by the referee we have combined the old panels of 5b-c in one graph (now Fig. 8a).

Minor comment 6: Line 183:184. It appears from the methods that trypsin was added after MAC formation, that should be mentioned:

Answer: We have now included this information in the Methods (page 16, line 483-484), results section (page 9, line 259), and figure legends (page 25, line 791-792).

Minor comment 7: Line 200-: AFM, briefly explain the difference between the "height" and the "phase" panels images to the non-expert

Answer: The results section now describes the terms 'height' and 'phase' (page 9, line 274-277). "Conv-MAC pores were further accentuated in the phase image (see Methods), which provides an alternative means to differentiate MAC pores from the underlying bacterial surface since it is sensitive to the local material properties (Fig. 8d, SFig. 5a)."

Minor comment 8: Suppl fig 6. Label the panels with the immobilization protocol used in each case

Answer: We have adapted the legends of Fig. 8c and d and SFig. 5 in the resubmitted manuscript to specify the immobilization protocol used in each case. In SFig. 5, we have also added symbols to the figure to clarify which immobilization protocol was used. Importantly, immobilization using the CellTak and PLL protocols were shown to yield similar results as can be observed in SFig. 5, where data was obtained using both protocols.

2nd Editorial Decision

28th November 2018

Thank you for submitting a revised version of your manuscript. It has now been seen by two of the original referees whose comments are shown below.

As you will see, while referee #3 finds that his/her concerns are sufficiently addressed and recommends the manuscript for publication, referee #4 points out that the proposed model for C5b6 assembly by C5 convertases should be carefully discussed in the context of the published literature.

In addition to resolving this remaining point from referee #4, there are a few editorial issues concerning text and figures that I need you to address before we can officially accept the manuscript.

REFeree REPORTS.

Referee #3:

The authors have addressed all my concerns and the revised manuscript is much improved. Overall, the findings are very exciting and they provide novel insight into how the membrane attack complex target and kill Gram negative bacteria.

Referee #4:

The revised manuscript is satisfactory in most aspects. However, the new figure 9 is very speculative and not supported by the newest model of MAC from Bubeck and coworkers cited as Ref 23 Menny et al directly stating: "While the core of C5b remains largely unchanged during the transition, C6 undergoes marked domain rearrangements upon integration into the MAC." and "Superposition of the soluble and membrane-associated forms of C5b6 show that although the relative orientation of the C5b thioester-like domain (TED) and C6 C-terminal complement control protein (CCP) domains remains unchanged....".

If the idea presented here that C5b may adopt a transient "C3b" like conformation with TED next to MG1 it would then have to return to the non-MG1 associated position now observed in both soluble C5b6 and in the MAC. This is not a very likely scenario. Menny et al also describe other aspects of MAC (open and closed MAC, glycan stabilization of MAC) that appears very relevant to the discussion in the present manuscript. There may be a problem in discussing such preprint data, but at least the revised manuscript should not propagate ideas like the C3b like position of the C5b that is clearly not supported by available structural data and data that is even quoted by the manuscript.

Minor points:

legend fig 3. Explain purpose of OmCI

legend fig 4d. perhaps "1% serum served as positive control"

Corresponding Author Name: Prof. S. Rooijackers

Journal Submitted to: The EMBO Journal

Manuscript Number: EMBOJ-2018-99852